

# Vertical profiles of aerosol optical properties and the solar heating rate estimated by combining sky radiometer and lidar measurements

Rei Kudo[1], Tomoaki Nshizawa[2], and Toshinori  Aoyagi[1]

[1]Meteorological Research Institute, Japan Meteorological Agency, Tsukuba, Japan

[2]National Institute for Environmental Studies, Tsukuba, Japan

*Correspondence to:* Rei Kudo (reikudo@mri-jma.go.jp)

**Abstract.** The SKYLIDAR algorithm was developed to estimate vertical profiles of aerosol optical properties from sky radiometer (SKYNET) and lidar (AD-Net) measurements. The solar heating rate was also estimated from the SKYLIDAR retrievals. The algorithm consists of two retrieval steps: (1) columnar properties are retrieved from the sky radiometer measurements and the vertically mean depolarization ratio obtained from the lidar measurements, and (2) vertical profiles are retrieved from the lidar measurements and the results of the first step. The derived parameters are the vertical profiles of the size distribution, refractive index (real and imaginary parts), extinction coefficient, single-scattering albedo, and asymmetry factor. Sensitivity tests were conducted by applying the SKYLIDAR algorithm to the simulated sky radiometer and lidar data for vertical profiles of three different aerosols, continental average, transported dust, and pollution aerosols. The vertical profiles of the size distribution, extinction coefficient, and asymmetry factor were well estimated in all cases. The vertical profiles of the refractive index and single-scattering albedo of transported dust were well estimated but not those of transported pollution aerosol. To demonstrate the performance and validity of the SKYLIDAR algorithm, we applied the SKYLIDAR algorithm to the actual measurements at Tsukuba, Japan. The detailed vertical structures of the aerosol optical properties and solar heating rate of transported dust and smoke were investigated. Examination of the relationship between the solar heating rate and the aerosol optical properties showed that the vertical profile of the asymmetry factor played an important role in creating vertical variation in the solar heating rate. We then compared the columnar optical properties between SKYLIDAR and SKYRAD.PACK retrievals, and the surface solar irradiance calculated from the SKYLIDAR retrievals was compared with pyranometer measurement. The results showed good agreements: The columnar values of the SKYLIDAR retrievals agreed with reliable SKYRAD.PACK retrievals, and the SKYLIDAR retrievals were sufficiently accurate to evaluate the surface solar irradiance.



# 1    Introduction

Aerosols significantly affect the Earth's radiation budget by scattering and absorbing incoming solar radiation (direct effect) and by modifying cloud droplet size and number density (indirect effect). In addition, solar heating of absorbing aerosols such as black carbon or dust affects the vertical profile of the temperature and the cloud cover (semi-direct effect; Hansen et al., 1997). To better understand these effects, it is essential to investigate the spatial and temporal variability of the microphysical and optical properties of aerosols, in particular, the vertical profiles of aerosol optical properties and solar heating of aerosols. The semi-direct effect depends on the vertical profiles of aerosols relative to cloud height (Koch and Del Genio, 2010). Under cloudless conditions, the vertical profile of solar heating of aerosols affects the evolution of the atmospheric boundary layer (Yu et al., 2002; Tsunematsu et al, 2006).

To evaluate solar heating of aerosols, vertical profiles of the extinction coefficient (loading), single-scattering albedo (ratio of scattering to scattering + absorption), and the phase function (or asymmetry factor, i.e., the asymmetry of forward and backward scattering) are necessary. The columnar properties of these parameters, but never their vertical profiles,  are routinely obtained by observational networks of sun-sky-scanning multi-wavelength photometers, such as AERONET (Holben et al., 1998) and SKYNET (Takamura and Nakajima, 2004; Nakajima et al., 2007). Active remote sensing by Mie lidar (MIEL) can provide vertical profiles of the extinction coefficient if the value of the extinction-to-backscatter ratio is assumed, but not those of the single-scattering albedo and phase function. Particle extinction and backscatter coefficients can be obtained by using Raman lidar (Ansman et al., 1992) and high-spectra-resolution lidar (HSRL; Shipley et al., 1983) measurements without any assumptions being necessary, and several studies have developed methods for estimating the vertical profiles of the aerosol size distribution and the real and imaginary parts of the refractive index from multi-wavelength Raman lidar data (Müller et al., 1999a, 1999b, 2000; Böckmann, 2001; Veselovskii et al., 2002). Then, on the basis of single-scattering theory, the extinction coefficient, single-scattering albedo, and phase function can be calculated from the size distribution and refractive index.

Synergistic algorithms that relate sun-sky-scanning photometer and MIEL data have been developed. The LiRIC (Lidar-Radiometer Inversion Code) algorithm (Chaikovsky et al., 2012; Wagner et al., 2013) uses the size distribution and refractive index of AERONET retrievals and estimates the vertical profiles of the fine and coarse modes of the size distribution from lidar data. Lopatin et al. (2013) developed the GARRLiC (Generalized Aerosol Retrieval from Radiometer and Lidar Combined data) algorithm by extending the LiRIC algorithm. GARRLiC separately estimates the columnar values of the refractive indices of the fine and coarse modes. Cuesta et al. (2008) developed the LidAlm (Lidar and Almucantar) algorithm, which estimates the vertical profiles after decomposing the AERONET size distribution into multiple log-normal modes. These algorithms provide good estimates of the aerosol vertical profiles from the lidar measurements, based on the AERONET retrievals.



SKYNET and AD-Net (Sugimoto et al., 2005) are dense observational networks of the sun-sky-scanning multi-wavelengths photometer (sky radiometer; SKYR) and MIEL, respectively, in the East Asian region, which is one of the world's major sources of dust and anthropogenic aerosols. The synergetic algorithms are useful for observing such aerosols. We developed a new algorithm, called SKYLIDAR, to estimate the vertical profiles of aerosol optical properties from the combination of SKYR and MIEL measurements. Similar to the above-mentioned synergetic algorithms, we assumed a bimodal size distribution but we challenged to estimate the vertical profile of the refractive index, which are sensitive to lidar measurements and necessary for determining single-scattering albedo and asymmetry factor. In addition, we estimated the solar heating rate from the SKYLIDAR retrievals.

The National Institute for Environment Studies (NIES) is developing the HSRL at 355 and 532 nm for updating AD-Net (Liu et al., 1999, 2002; Nishizawa et al., 2012). For the future SYNET and AD-Net networks, we designed the SKYLIDAR algorithm so that it could be applied to both MIEL and HSRL measurements, and the algorithm was tested using the simulated HSRL data in this study.

We describe the SKYR, MIEL, and HSRL measurements, the SKYLIDAR algorithm, and the procedure for calculating the solar heating rate in Sect. 2. In Sect. 3, we evaluate the performance of the SKYLIDAR algorithm by sensitivity tests performed with simulated SKYR, MIEL, and HSRL data for three different vertical profiles of aerosols. In Sect. 4, we apply the algorithm to actual SKYR and MIEL measurements obtained at Tsukuba, Japan, estimating the optical properties and the solar heating rate of transported dust and smoke. We also compare the columnar optical properties of the SKYLIDAR retrievals during 2012 and 2013 with SKYRAD.PACK retrievals, and we compare surface solar irradiances calculated from the SKYLIDAR retrievals with those measured by pyranometer. In Sect. 5, we summarize the results.

## 2    Method

The SKYLIDAR algorithm uses the solar direct and diffuse radiations measured by the sun photometer SKYR in the SKYNET and the attenuated backscatter coefficient and the total depolarization ratio by the lidar MIEL in the AD-Net. In addition, the algorithm can be applicable to the lidar HSRL, which being developed by the NIES. The details of these instruments are described in the beginning of this section.

The SKYLIDAR algorithm (Fig. 1) consists of two retrieval steps. In step 1 (Fig. 1a), the columnar microphysical and optical properties of aerosols are estimated from SKYR measurements and the vertically mean depolarization ratio obtained from MIEL measurements. In step 2 (Fig. 1b), the vertical profiles of the microphysical and optical properties are estimated from MIEL (and HSRL) measurements and the columnar properties obtained in step 1.



## 2.1 SKYLIDAR algorithm

### 2.1.1 Sky radiometer and Lidar measurements

The SKYR (Prede Co., Ltd, Tokyo, Japan), deployed in the SKYNET, is a scanning photometer that measures direct solar radiation and the angular distributions of the diffuse radiation in solar almucantar or principal plane geometries at wavelengths of 315, 340, 380, 400, 500, 675, 870, 940, 1020, 1627, and 2200 nm. Aerosol optical thickness is obtained from the direct solar radiation measurement by using a calibration constant determined by the improved langley method (Nakajima et al., 1996; Tanaka et al., 1986). Diffuse radiance is obtained from the measured diffuse radiation and the field of view (solid view angle), which is determined by scanning the distribution of radiation around the solar disk (Nakajima et al., 1996). Our algorithm uses aerosol optical thickness $\tau_{ext}(\lambda)$ and the diffuse radiance normalized by the direct solar radiation $I(\Theta, \lambda)$ at wavelength $\lambda$ and at scattering angle $\Theta$ in the solar almucantar geometry (Fig. 2). The normalized diffuse radiance $I(\Theta, \lambda)$ is defined as

$$I(\Theta, \lambda) \equiv I(\theta_0, \phi, \lambda) = \frac{F_{dif}(\theta_0, \phi, \lambda)}{F_{dir}(\lambda) m \Delta\Omega(\lambda)}, \qquad (1)$$

$$\cos\Theta = \cos^2\theta_0 + \sin^2\theta_0 \cos(\phi), \qquad (2)$$

where $\theta_0$ is solar zenith angle, $\phi$ is the observation azimuth angle ( zero at the solar azimuth angle), $F_{dif}(\theta_0, \phi, \lambda)$ is the diffuse radiation in the solar almucantar geometry, $m = 1/cos\theta_0$ is the optical air mass, $\Delta\Omega(\lambda)$ is solid view angle, and $F_{dir}(\lambda)$ is the direct solar radiation. The wavelength $\lambda$ used in the algorithm is 340, 380, 400, 500, 675, 870, and 1020 nm. The scattering angle $\Theta$ used in the algorithm is 3, 4, 5, 7, 10, 15, 20, 25, 30, 40, 50, 60, 70, 80, 90, 100, 110, 120, 130, 140, 150, and 160º in the solar almucantar plane. Note that the maximum value of the scattering angle depends on the solar zenith angle.

Diffuse radiances scanned in the almucantar geometry are suitable for estimating columnar properties because the effect of the aerosol vertical profile to diffuse radiances on the almucantar plane is weak (Torres et al., 2013). Because the SKYR measures both direct and diffuse radiation with the same detector, the calibration constant cancels out by normalizing diffuse radiance to direct radiation, and the calibration uncertainty is neglected in $I(\Theta, \lambda)$.

The SKYLIDAR algorithm can be applied to both MIEL and HSRL measurements developed by NIES. We use the attenuated backscatter coefficients for total (particulate + molecular) scattering at 532 and 1064 nm, and the total





depolarization ratio at 532 nm in MIEL measurements. These data are routinely calibrated (Shimizu et al., 2004, 2010). The HSRL measurements are the attenuated backscatter coefficients for molecular scattering at 355 and 532 nm.

The lidar signals from near the surface contain errors owing to the incomplete overlap between the transmitted laser beams and the receiver field of view. We therefore excluded the original MIEL and HSRL data from below the altitude of 300 m and used data extrapolated linearly from measurements obtained above 300 m in the algorithm.

In many lidar retrievals, the attenuated backscatter coefficient is normalized by the attenuated backscatter coefficient at a reference altitude, where aerosol is neglected. However, the reference altitude is usually high and the reference attenuated backscatter coefficient is influenced by the large noise. Therefore, we normalized the attenuated backscatter coefficients for total and molecular scattering by their vertical means from the surface to top of aerosol layer. The top altitude is determined by the empirically determined threshold of the MIEL backscatter signal at 1064 nm. Because the random noise included in the vertically mean attenuated backscatter coefficient is expected to be much smaller than particulate + molecular backscatter, the calibration constant of the attenuated backscatter coefficient cancels out.

Thus, the SKYLIDAR algorithm can be applied to the SKYR data ($\tau_{ext}(\lambda)$, $I(\Theta, \lambda)$) and the following three data sets of MIEL and HSRL:

(Type 1) $\beta_{MIE}(\lambda, z)$ at $\lambda = 532$ and 1064 nm, and $\delta(\lambda, z)$ at $\lambda = 532$ nm,

(Type 2) $\beta_{MIE}(\lambda, z)$ at $\lambda = 532$ and 1064 nm, $\delta(\lambda, z)$ at $\lambda = 532$ nm, and $\beta_{RAY}(\lambda, z)$ at $\lambda = 532$ nm,

(Type 3) $\beta_{MIE}(\lambda, z)$ at $\lambda = 355$, 532, and 1064 nm, $\delta(\lambda, z)$ at $\lambda = 532$ nm, and $\beta_{RAY}(\lambda, z)$ at $\lambda = 355$ and 532 nm,

where $z$ is altitude, $\lambda$ is wavelength, $\beta_{MIE}(\lambda, z)$ is the normalized attenuated backscatter coefficient for total scattering and $\delta(\lambda, z)$ is the total depolarization ratio, and $\beta_{RAY}(\lambda, z)$ is the normalized attenuated backscatter coefficient for molecular scattering. Type 1 data set is only MIEL measurements, and Type 2 and 3 data sets include HSRL measurements.

### 2.1.2    Step 1

The columnar properties of aerosols are estimated from the SKYR measurements ($\tau_{ext}^{mea}(\lambda)$, $I^{mea}(\Theta, \lambda)$) and the depolarization ratio averaged from the surface to the top of the aerosol layer ($\delta_{ave}^{mea}(532$ nm$)$). The aerosol parameters to be estimated are the real and imaginary parts of the refractive index at SKYR wavelengths, the volume size distribution, and the volume ratio of non-spherical particles to total particles in the coarse mode.

To estimate the aerosol parameters, our algorithm uses the maximum likelihood method. The aerosol parameters for the best fit to all of the input data are obtained by maximizing the probability density function:





$$P(\mathbf{y(x)}|\mathbf{y}^{mea}) \propto \exp[-\tfrac{1}{2}(\mathbf{y(x)} - \mathbf{y}^{mea})^T(\mathbf{W}^2)^{-1}(\mathbf{y(x)} - \mathbf{y}^{mea})$$

$$-\tfrac{1}{2}\mathbf{y}_a(\mathbf{x})^T(\mathbf{W}_a^2)^{-1}\mathbf{y}_a(\mathbf{x})], \tag{3}$$

where vector $\mathbf{y}^{mea}$ describes the measurements, vector $\mathbf{x}$ describes the aerosol parameters to be estimated, vector $\mathbf{y(x)}$ comprise the values corresponding to $\mathbf{y}^{mea}$ calculated by the forward model, and vector $\mathbf{y}_a(\mathbf{x})$ comprises the a priori constraints on $\mathbf{x}$. The matrix $\mathbf{W}$ is the covariance matrix of $\mathbf{y}$ and is assumed to be diagonal in this study. The diagonal elements of $\mathbf{W}$ are the errors of each measurement. Matrix $\mathbf{W}_a$ comprises the weights that determine the strength of the a priori constraints. The maximum of $P(\mathbf{y(x)}|\mathbf{y}^{mea})$ is obtained by minimizing the objective function,

$$f(\mathbf{x}) = \tfrac{1}{2}(\mathbf{y(x)} - \mathbf{y}^{mea})^T(\mathbf{W}^2)^{-1}(\mathbf{y(x)} - \mathbf{y}^{mea}) + \tfrac{1}{2}\mathbf{y}_a(\mathbf{x})^T(\mathbf{W}_a^2)^{-1}\mathbf{y}_a(\mathbf{x}). \tag{4}$$

We search for the best $\mathbf{x}$, which minimizes $f(\mathbf{x})$, by iterations of the Gauss-Newton method with a line search, $\mathbf{x}_{i+1} = \mathbf{x}_i + \alpha_j \mathbf{d}_i$ ("Update $\mathbf{x}$" in Fig. 1). This minimization procedure is described in Sects 2.1.4 and 2.1.5.

In step 1, the vector $\mathbf{y}^{mea}$ consists of the SKYR and MIEL measurements and is written as

$$\mathbf{y}^{mea} = (\cdots \quad \tau_{ext}^{mea}(\lambda) \quad \cdots \quad \cdots \quad I^{mea}(\Theta, \lambda) \quad \cdots \quad \delta_{ave}^{mea}(532 \text{ nm})). \tag{5}$$

Vector $\mathbf{x}$ describes the aerosol parameters and is defined as

$$\mathbf{x} = (\cdots \quad n(\lambda) \quad \cdots \quad \cdots \quad k(\lambda) \quad \cdots \quad C_1 \quad C_2 \quad r_{m,1} \quad r_{m,2} \quad s_1 \quad s_2 \quad \varepsilon), \tag{6}$$

where $n(\lambda)$ and $k(\lambda)$ are the real and imaginary parts of the refractive index at SKYR wavelengths, and $C_1, C_2, r_{m,1}, r_{m,2}, s_1$, and $s_2$ are parameters of the bi-modal lognormal size distribution,



$$\frac{dV(r)}{d\ln r} = \sum_{i=1}^{2} \frac{dV_i(r)}{d\ln r} = \sum_{i=1}^{2} \frac{C_i}{\sqrt{2\pi}s_i} \exp\left[-\frac{1}{2}\left(\frac{\ln r - \ln r_{m,i}}{s_i}\right)^2\right],$$ (7)

where $C_i$, $r_{m,i}$, and $s_i$ are volume, radius, and width, respectively of the fine ($i = 1$) and coarse ($i = 2$) modes. $\varepsilon$ is the volume ratio of non-spherical particles to total particles in the coarse mode.

We constructed the forward models $\mathbf{y}(\mathbf{x})$ to calculate $\tau_{ext}(\lambda)$, $I(\Theta, \lambda)$, and $\delta_{ave}(532\text{nm})$ from the above-mentioned aerosol parameters. The optical properties of aerosols were calculated by a method similar to that of Lopatin et al. (2013) as follows:

$$\tau_{\frac{ext}{sca}}(\lambda) = \sum_k \frac{dV_1(r_k)}{d\ln r} K^S_{\frac{ext}{sca}}(\lambda, n, k, r_k) + \sum_k (1-\varepsilon) \frac{dV_2(r_k)}{d\ln r} K^S_{\frac{ext}{sca}}(\lambda, n, k, r_k)$$

$$+ \sum_k \varepsilon \frac{dV_2(r_k)}{d\ln r} K^{NS}_{ext/sca}(\lambda, n, k, r_k),$$ (8)

$$\tau_{sca}(\lambda) P_{ii}(\Theta, \lambda) = \sum_k \frac{dV_1(r_k)}{d\ln r} K^S_{ii}(\Theta, \lambda, n, k, r_k)$$
$$+ \sum_k (1-\varepsilon) \frac{dV_2(r_k)}{d\ln r} K^S_{ii}(\Theta, \lambda, n, k, r_k) + \sum_k \varepsilon \frac{dV_2(r_k)}{d\ln r} K^{NS}_{ii}(\Theta, \lambda, n, k, r_k),$$ (9)

where $\tau_{ext/sca}(\lambda)$ denotes the optical thickness for extinction and scattering, and $\tau_{sca}(\lambda)P_{ii}(\Theta, \lambda)$ denotes the directional scattering corresponding to the scattering matrix elements $P_{ii}(\Theta, \lambda)$. $K^S_{\dots}$ and $K^{NS}_{\dots}$ are the kernels of extinction and scattering properties for spherical and non-spherical particles, respectively. The kernel for spherical particles was constructed by using Mie theory. The kernel for non-spherical particles was constructed by using the data table of Dubovik et al. (2006), which assumes randomly oriented polydisperse spheroids with a fixed aspect ratio distribution for mineral dust.

$I(\Theta, \lambda)$ is computed by the radiative transfer code in the SKYRAD.PACK ver. 4.2 (Nakajima et al., 1996). Although only one atmospheric layer is considered by SKYRAD.PACK, we assumed that the atmosphere consists of two layers. The bottom layer includes aerosols, and its top altitude $z_{max}$ is determined by lidar measurement. The upper layer is aerosol-free. The Rayleigh scattering is calculated by the method of Bucholtz (1995). The vertical ozone profile is approximated by the formula of Green (1964), and the ozone absorption coefficient is adopted from the LOWTRAN 7 database (Kneizys et al., 25   1988).

$\delta_{ave}(\lambda)$ is calculated as





$$\delta_{ave}(\lambda) = \left(\frac{\beta_m(\lambda)\delta_m(\lambda)}{1+\delta_m(\lambda)} + \frac{\beta_p(\lambda)\delta_p(\lambda)}{1+\delta_p(\lambda)}\right) \Big/ \left(\frac{\beta_m(\lambda)}{1+\delta_m(\lambda)} + \frac{\beta_p(\lambda)}{1+\delta_p(\lambda)}\right),\tag{10}$$

$$\beta_{p/m}(\lambda) = \tau_{sca,p/m}(\lambda)P_{11,p/m}(180^\circ,\lambda),\tag{11}$$

$$\delta_{p/m}(\lambda) = \frac{1-P_{22,p/m}(180^\circ,\lambda)/P_{11,p/m}(180^\circ,\lambda)}{1+P_{22,p/m}(180^\circ,\lambda)/P_{11,p/m}(180^\circ,\lambda)},\tag{12}$$

where $\beta_{p/m}(\lambda)$, $\delta_{p/m}(\lambda)$, and $\tau_{sca,p/m}(\lambda)$ are the backscatter coefficient, the depolarization ratio, and the scattering optical thickness, respectively, for particulate ($p$) and molecular ($m$) scattering in the bottom aerosol layer.

The first term of Eq. (4) is calculated by using the above-mentioned $\mathbf{y}^{mea}$ and $\mathbf{y}(\mathbf{x})$. The values of the diagonal matrix $\mathbf{W}$ are the measurement errors, which were assumed to be 0.01 for $\tau_{ext}^{mea}(\lambda \geq 500\text{nm})$ and 0.02 for $\tau_{ext}^{mea}(\lambda < 500\text{nm})$, 5 % for $I^{mea}(\Theta,\lambda)$, and 20 % for $\delta_{ave}^{mea}(532\,\text{nm})$.

We introduced a priori smoothness constraints for the wavelength dependencies of the refractive index by using the method of Dubovik and King (2000) in order to reduce the effects of measurement errors on retrievals. The first derivatives of the

refractive index with respect to the wavelengths are defined as

$$\mathbf{y}_a(\mathbf{x}) = \left(\cdots \quad \frac{\ln n(\lambda_i)-\ln n(\lambda_{i+1})}{\ln\lambda_i-\ln\lambda_{i+1}} \quad \cdots \quad \cdots \quad \frac{\ln k(\lambda_i)-\ln k(\lambda_{i+1})}{\ln\lambda_i-\ln\lambda_{i+1}} \quad \cdots\right).\tag{13}$$

The second term in Eq. (4) is calculated by Eq. (13). The values entered in the weight matrix $\mathbf{W}_a$ were 0.2 for the real part

and 1.25 for the imaginary part. These values are used in the AERONET retrieval (Dubovik and King, 2000) for constraining the spectral variability of the refractive index to some practically reasonable ranges.

The objective function (Eq. (4)) is minimized by the procedures described in Sects 2.1.4 and 2.1.5, and the columnar properties of the real and imaginary parts of the refractive index ($n^{s1}(\lambda)$ and $k^{s1}(\lambda)$) at SKYR wavelengths, the size distribution ($C_{1,2}^{s1}$, $r_{1,2}^{s1}$, and $s_{1,2}^{s1}$), and the volume ratio of non-spherical particles to total particles in the coarse mode ($\varepsilon^{s1}$) are

optimized. The aerosol optical thickness ($\tau_{ext}^{s1}(\lambda)$) and single-scattering albedo ($\omega_0^{s1}(\lambda)$) at SKYR wavelengths are



calculated with Equations (7) to (9). $n^{s1}(\lambda)$, $k^{s1}(\lambda)$, $\tau_{ext}^{s1}(\lambda)$, and $\omega_0^{s1}(\lambda)$ at MIEL wavelengths are calculated by interpolation and extrapolation. These step 1 results are input to step 2 (see Figs. 1a and b).

### 2.1.3   Step 2

The vertical profiles of the refractive index at MIEL wavelengths, the size distribution, and the volume ratio of the non-spherical particles to total particles in the coarse mode are optimized to MIEL (and HSRL) measurements and the columnar properties obtained in step 1 by the same strategy. The final outputs of the extinction coefficients, single-scattering albedo, and asymmetry factor at MIEL wavelengths are calculated from the optimized aerosol parameters.

The columnar properties input from step 1 are the aerosol optical thickness $\tau_{ext}^{s1}(\lambda)$ and the single-scattering albedo $\omega_0^{s1}(\lambda)$ at MIEL wavelengths. The MIEL and HSRL measurements are three data sets described in Sect. 2.1.1. $\mathbf{y}^{mea}$ for Type 1 data set is defined as

$$
\begin{aligned}
\mathbf{y}^{mea} = (\cdots \quad \tau_{ext}^{s1}(\lambda) \quad \cdots \quad \cdots \quad \omega_0^{s1}(\lambda) \quad \cdots \quad \cdots \quad \beta_{MIE}^{mea}(\lambda, z) \quad \cdots \\
\cdots \quad \delta^{mea}(\lambda, z) \quad \cdots).
\end{aligned}
\tag{14}
$$

$\mathbf{y}^{mea}$ for Type 2 and 3 data sets is defined as

$$
\begin{aligned}
\mathbf{y}^{mea} = (\cdots \quad \tau_{ext}^{s1}(\lambda) \quad \cdots \quad \cdots \quad \omega_0^{s1}(\lambda) \quad \cdots \quad \cdots \quad \beta_{MIE}^{mea}(\lambda, z) \quad \cdots \\
\cdots \quad \delta^{mea}(\lambda, z) \quad \cdots \quad \cdots \quad \beta_{RAY}^{mea}(\lambda, z) \quad \cdots).
\end{aligned}
\tag{15}
$$

The aerosol parameter $\mathbf{x}$ is defined as

$$
\begin{aligned}
\mathbf{x} = (\cdots \quad n(\lambda, z) \quad \cdots \quad \cdots \quad k(\lambda, z) \quad \cdots \\
\cdots \quad C_1(z) \quad \cdots \quad \cdots \quad C_2(z) \quad \cdots \quad \cdots \quad \varepsilon(z) \quad \cdots).
\end{aligned}
\tag{16}
$$





The bi-modal size distribution (Eq. (7)) is also used in step 2, but the mode radii and the widths of the fine and coarse modes are fixed by the columnar values obtained in step 1 ($r_{1,2}^{s1}$, and $s_{1,2}^{s1}$).

In the forward model $\mathbf{y}(\mathbf{x})$ of step 2, the aerosol optical properties at each altitude are calculated with Equations (7) to (9), but note that the extinction/scattering coefficients are calculated.

The normalized attenuated backscatter coefficients for total and molecular scattering are calculated by the lidar equations,

$$\beta_{MIE}(\lambda, z) = \left(\beta_m(\lambda, z) + \beta_p(\lambda, z)\right)$$

$$\exp\left(-2\int_0^z \sigma_{ext,p}\left(\lambda, z^{'}\right) + \sigma_{ext,m}\left(\lambda, z^{'}\right) dz^{'}\right)/\beta_{MIE,ave}(\lambda), \tag{17}$$

$$\beta_{RAY}(\lambda, z) = \beta_m(\lambda, z)\exp\left(-2\int_0^z \sigma_{ext,p}\left(\lambda, z^{'}\right) + \sigma_{ext,m}\left(\lambda, z^{'}\right) dz^{'}\right)/\beta_{RAY,ave}(\lambda),$$

$$\tag{18}$$

where $\sigma_{ext,p/m}$ are the extinction coefficients for particulate ($p$) and molecular ($m$) scattering, and $\beta_{MIE,ave}(\lambda)$ and $\beta_{RAY,ave}(\lambda)$ are the vertical means of the calculated attenuated backscatter coefficients. The total depolarization ratio

($\delta(\lambda, z)$) at each altitude is calculated with Eqs. (10) to (12).

The first term of Eq. (4) is calculated with the above-mentioned $\mathbf{y}^{mea}$ and $\mathbf{y}(\mathbf{x})$. The values of the diagonal matrix $\mathbf{W}$ were assumed to be 0.01 for $\tau_{ext}^{s1}(\lambda \geq 532nm)$, 0.02 for $\tau_{ext}^{s1}(\lambda < 532nm)$, 0.05 for $\omega_0^{s1}(\lambda)$, 10 % for $\beta_{MIE}^{mea}(\lambda, z)$, 15 % for $\beta_{RAY}^{mea}(\lambda, z)$, and 20 % for $\delta^{mea}(\lambda, z)$.

In step 2, the number of estimated parameters is larger than the number of measurements, so the lidar measurements would
be insufficient for retrieving unique solutions of the refractive index even if the columnar properties obtained in step 1 are added to $\mathbf{y}^{mea}$. Therefore, we added a priori distribution constraints to the refractive index. We assumed that the values of the real and imaginary parts were approximately those obtained in step 1 and constrained their values by zeroing the function,

$$\mathbf{y}_a(\mathbf{x}) = (\cdots \quad \ln n(\lambda, z) - \ln n^{s1}(\lambda) \quad \cdots \quad \cdots \quad \ln k(\lambda, z) - \ln k^{s1}(\lambda) \quad \cdots),$$

$$\tag{19}$$





where $n^{s1}(\lambda)$ and $k^{s1}(\lambda)$ are the real and imaginary parts obtained in step 1. The weight $\mathbf{W}_a$ in Eq. (4), which determines the strength of the constraints, is obtained by a method similar to that of Dubovik and King (2000). The possible variability ranges of the refractive index for aerosols are from 1.33 to 1.6 for the real part, and from 0.0005 to 0.5 for the imaginary part. We considered these intervals to be 68 % confidence intervals, $[\ln n^{s1} - \Delta n, \ln n^{s1} + \Delta n]$ and $[\ln k^{s1} - \Delta k, \ln k^{s1} + \Delta k]$,
and determined the weight values as

$$W_a = \begin{cases} 0.5(\ln n_{max} - \ln n_{min}), & \text{for real part} \\ 0.5(\ln k_{max} - \ln k_{min}), & \text{for imaginary part} \end{cases}. \tag{20}$$

The objective function is minimized by the procedures described in Sects 2.1.4 and 2.1.5, and the vertical profiles of the real
and imaginary parts of the refractive index $(n(\lambda, z), k(\lambda, z))$ at MIEL wavelengths, the size distribution $(C_1(z), C_2(z))$, and the volume ratio of non-spherical particles to total particles in the coarse mode $(\varepsilon(z))$ are optimized. Finally, the vertical profiles of the refractive index, size distribution $(\frac{dV(r,z)}{d\ln r})$, extinction coefficients $(\sigma_{ext}(\lambda, z))$, single-scattering albedo $(\omega_0\,(\lambda, z))$, and asymmetry parameter $(g(\lambda, z))$ are output. The wavelengths of these optical properties are 532 and 1064 nm for Type 1 and 2 data sets, and 355, 532, 1064 nm for Type 3 data set.

**2.1.4   Minimization procedure**

In both steps 1 and 2, $\mathbf{x}$ was optimized to $\mathbf{y}^{mea}$ by finding the minimum of $f(\mathbf{x})$ in $\mathbf{x}$ space. We employed the Gauss-Newton method to search for the minimum. The Gauss-Newton method searches for the minimum by iteratively updating $\mathbf{x}$, and it is usually combined with a line search method (Nocedal and Wright, 2006). In this procedure (Fig. 1c), $\mathbf{x}$ is updated by $\mathbf{x}_{i+1} = \mathbf{x}_i + \alpha_j \mathbf{d}_i$, where the value of $f(\mathbf{x})$ in the $\mathbf{x}$ space decreases in the vector $\mathbf{d}_i$ direction, and $\alpha_j$ is a positive parameter
that minimizes $f(\mathbf{x})$ in direction $\mathbf{d}_i$. $\mathbf{d}_i$ is determined by the Gauss-Newton method in the outer-loop, and $\alpha_j$ is determined by a line search in the inner loop. $\mathbf{d}_i$ is obtained by solving the normal equation,

$$[\mathbf{J}(\mathbf{x}_i)^T(\mathbf{w}^2)^{-1}\mathbf{J}(\mathbf{x}_i) + (\mathbf{w}_a^2)^{-1}]\mathbf{d}_i = -\mathbf{J}(\mathbf{x}_i)^T(\mathbf{w}^2)^{-1}(\mathbf{y}(\mathbf{x}_i) - \mathbf{y}^{mea})$$
$$+(\mathbf{w}_a^2)^{-1}\mathbf{y}_a(\mathbf{x}_i), \tag{21}$$



where $\mathbf{J}(\mathbf{x}_i)$ is the Jacobi matrix and is calculated as the first derivatives of $\mathbf{y}(\mathbf{x}_i)$ in the near vicinity of $\mathbf{x}_i$. We solved this normal equation by Singular Value Decomposition (Press et al., 1992). After $\mathbf{d}_i$ is determined, $\alpha_j$ is searched for by the iteration of $\alpha_{j+1} = \eta \alpha_j$. The initial value of $\alpha_j$ is 1.0, and the value of $\eta$ is set to 0.5. $\alpha_j$ is iteratively decreased until the Armijo condition is satisfied,

$$f\left(\mathbf{x}_i + \alpha_j \mathbf{d}_i\right) \le f(\mathbf{x}_i) + \gamma \alpha_j \nabla f(\mathbf{x}_i)^T \mathbf{d}_i, \, 0 < \gamma < 1, \tag{22}$$

where $\gamma$ is an arbitrary constant that we set to 0.001. This line search inhibits unstable oscillation in the Gauss-Newton method by limiting the length of $\|\mathbf{x}_{i+1} - \mathbf{x}_i\|$; as a result monotonic and stable convergence is obtained.

10   ### 2.1.5   Logarithmic transformation

In our minimization problem, the number of elements in $\mathbf{x}$ is on the order of $10^2$ in step 2. Furthermore, the elements in $\mathbf{x}$ and $\mathbf{y}$ have different units and values that vary over a wide range of magnitude. Under these conditions, there are too many iterations of the Gauss-Newton method, and convergence cannot be obtained. It is therefore necessary to make the convergence efficient. Dubovik and King (2000) developed a logarithmic transformation technique for $\mathbf{y}$ and $\mathbf{x}$ by which

15   $f(\mathbf{x})$ becomes dimensionless, because the term $\mathbf{y}(\mathbf{x}) - \mathbf{y}^{mea}$ in Eq. (4) is expressed as $\ln(\mathbf{y}(\mathbf{x})/\mathbf{y}^{mea})$. This makes it simple to operate simultaneously with $\mathbf{y}$ that has different units and values. Furthermore, the logarithmic transformation prevents $\mathbf{x}$ from assuming a negative value. We developed more effective transformation techniques. $y$ is transformed as

$$Y = \ln(y - y_{min}), \tag{23}$$

where $y_{min}$ is a possible minimum value of $y$. When the aerosol load is small, the attenuated backscatter coefficient and depolarization ratio measurements can have negative values because of the large random noise. Although the logarithmic transformation of Dubovik and King (2000) cannot be applied to negative values, Eq. (23) can.

$x$ is transformed by

$$X = \ln\left(\frac{x - x_{min}}{x_{max} - x}\right), \tag{24}$$



where $x_{min}$ and $x_{max}$ are minimum and maximum possible values of $x$, respectively. This equation can be inverted as

$$x = \frac{x_{min} + x_{max}\exp(X)}{1 + \exp(X)}.$$

(25)

The value of $x$ is sometimes limited for physical or numerical reasons. For example, for rapid computation, we usually construct a look-up table of optical properties with minimum and maximum values of the refractive index. If the refractive index exceeds its maximum or minimum values in the retrieval process, then the optical properties cannot be calculated and the retrieval process stops. The transformation by Eq. (24) prevents $x$ from exceeding its minimum or maximum limitations.

Furthermore, because X and Y are dimensionless, it is simpler to deal with the multiple parameters simultaneously. The transformations of the real and imaginary parts of the refractive index by Eq. (24) are illustrated in Fig. 3. Although the magnitude of each part varies over a different wide range, the magnitudes of the transformed values vary over the same range. However, the value of $x - x_{min}$ or $x_{max} - x$ sometimes become zero owing to a rounding error when the value of $x$ is close to $x_{min}$ or $x_{max}$. In this case, Eq. (24) cannot be calculated. Therefore, Eq. (24) should be computed carefully.

By applying the above transformations, the problem of searching for the minimum value of $f(\mathbf{x})$ in $\mathbf{x}$ space becomes a search for the minimum in dimensionless $\mathbf{X}$ space. This transformation has a scaling effect. In minimization problems, the convergence rate of an algorithm becomes more rapid if the problem is well scaled (Nocedal and Wright, 2006). An example of poor scaling is the function, $f(x_1, x_2) = 10^{10}x_1^2 + x_2$, which is sensitive to small changes in $x_1$ but not sensitive to $x_2$. If we define a new variable $X_1 = 10^5 x_1$ and minimize the function in terms of $X_1$ and $x_2$, the optimum values of $X_1$ and $x_2$ can

be found more rapidly. The condition number, which is a measure of the scaling, is defined as the ratio of the maximum to the minimum singular value of matrix $[\mathbf{J}(\mathbf{x}_i)^T (\mathbf{w}^2)^{-1} \mathbf{J}(\mathbf{x}_i) + (\mathbf{w}_a^2)^{-1}]$ of Eq. (21) in this study. If the condition number is close to 1, the problem is well scaled. The condition number for step 2 in this study is on the order of $10^{19}$ for the case without any transformations, $10^4$ when the logarithmic transformation of Dubovik and King (2000) is applied, and $10^2$ when our transformations are applied. In actuality, no step 2 results can be obtained unless transformations are applied.

## 2.2 Solar heating rate

The vertical profiles of the solar heating rate are calculated from the vertical profiles of the extinction coefficient, single-scattering albedo, and phase function in the broadband wavelength regions of solar radiation by using the radiative transfer model. However, the only wavelengths of the optical properties obtained by the SKYLIDAR algorithm are MIEL





wavelengths, i.e., 532 and 1064 nm for Type 1 data set. We calculated the refractive index between 532 and 1064 nm by interpolation. For wavelengths less than 532 nm and greater than 1064 nm, the refractive index at 532 and 1064 nm were used. The extinction coefficient, single-scattering albedo, and phase function in the broadband wavelength regions were calculated from the linearly estimated refractive index and the size distribution. The radiative transfer calculation was performed with our developed code (Asano and Shiobara, 1989; Nishizawa et al., 2004; Kudo et al., 2011). The solar spectrum between 300 and 3000 nm was divided into 54 intervals. Gaseous absorption by water vapor, carbon dioxide, oxygen, and ozone were included in the radiative transfer model.

## 3 Sensitivity tests using simulated data

To evaluate the performance of the SKYLIDAR algorithm, we conducted sensitivity tests using simulated SKYR, MIEL, and HSRL data (Type 1 to 3 data sets). Three aerosol vertical profile patterns were used for the simulation: (1) continental average, (2) continental average + transported dust in the upper air, and (3) continental average + transported pollution aerosol in the upper air. The microphysical and optical properties, and the vertical profiles of the continental average, transported dust, and transported pollution aerosols are summarized in Table 1. The continental average was defined as an external mixture of water-soluble particles, soot particles, and insoluble particles (Hess et al., 1999). The pollution aerosol in this sensitivity test was defined as an external mixture of water-soluble and soot particles. The kernels of the non-spherical particles described in Sect. 2.1.2 were used for calculating the optical properties of the insoluble and dust particles. The sensitivity tests were conducted for aerosol optical thickness of 0.05, 0.1, 0.3, 0.5, 0.8, and 1.2 at 500 nm.

Figure 4 illustrates the retrieval results from the simulated data for the continental average aerosol with the aerosol optical thickness of 0.05 at 500nm. The retrieval results with and without HSRL data were the same. The estimated real and imaginary parts of the refractive index were almost constant, and no vertical variations were estimated. Although the coarse mode of the size distribution was overestimated at low altitudes, overall the vertical profiles of the size distribution were estimated well. The vertical profiles of the extinction coefficients, the single-scattering albedo, and the asymmetry factor were also reproduced well.

We also conducted sensitivity tests using simulated data with random errors to investigate the performance of the algorithm under more realistic conditions. The given random errors were 2 % for direct solar radiation, 3 % for diffuse sky radiances, 5 % for the attenuated backscatter coefficient for total scattering, 10 % for the attenuated backscatter coefficient for molecular scattering, and 15 % for the total depolarization ratio. Figure 5 illustrates the retrieval results from the simulated data for the continental average aerosol, but the simulated data includes random errors. The vertical profile of the extinction coefficient was estimated well. However, the estimated single-scattering albedo and asymmetry factor exhibited large oscillations, so their vertical profiles were not clear.





Figure 6 presents the retrieval results from the simulated data for the transported dust with the aerosol optical thickness of 0.5 at 500 nm. The vertical profiles of the refractive index, other than the real part at 355 and 532 nm, were well estimated. The vertical profiles of the size distributions were also estimated well. The vertical profiles of the extinction coefficients, the single-scattering albedo, and the asymmetry factor were reproduced well. Figure 7 is the retrieval results from the simulated

data with random errors. There were small oscillations in the vertical profiles of all the retrievals but the results were almost same as those in Fig. 6.

Figure 8 plotted the retrieval results from the simulated data for the transported pollution aerosol with the aerosol optical thickness of 0.3 at 500nm. The vertical profile of the size distribution was estimated well, but the vertical profiles of the refractive index were not. In this test, the large values of the imaginary part of the refractive index and the small values of

the single-scattering albedo at upper altitudes were important characteristics that were not reproduced even when HSRL data was used in the retrieval. Figure 9 is the retrieval results from the simulated data with random errors. All the vertical profiles had small oscillations but were almost same as those in Fig. 8.

Overall in these tests, our algorithm estimated well the vertical profiles of the size distribution. Therefore, the vertical profiles of the extinction coefficient and the asymmetry factor were reproduced well. The vertical profiles of the refractive

index and the single-scattering albedo of transported dust were also estimated well, but not those of transported pollution aerosol. These characteristics were consistently observed regardless of the aerosol optical thickness value of the simulated data.

The retrieval results obtained with and without HSRL data did not differ. The advantage of HSRL data is that particle backscatter and extinction coefficients are obtained separately. Manipulating the MIEL data together with the aerosol optical

thickness would have an effect similar to the addition of HSRL data. In this regard, our algorithm cannot utilize HSRL data; thus further development of the algorithm is necessary.

The random errors had a large influence when the aerosol optical thickness was small (Fig. 5). The estimated single-scattering albedo and asymmetry factor exhibited large oscillations, so their vertical profiles were not clear. However, when the optical thickness was larger, the influence of the random error was small (Figs 7 and 9).

**4      Application to observational data**

We applied our developed SKYLIDAR algorithm to actual SKYR and MIEL measurements obtained during 2012 and 2013 at Tsukuba ($140.12^{o}$E, $36.05^{o}$N), Japan.



## 4.1    Application of the SKYLIDAR algorithm to actual measurements

The retrieval results of the aerosol optical properties and the solar heating rate for transported dust observed in 2 April, 2012 are shown in Fig. 10. High values of the extinction coefficients were observed in layer 1 (1-2 km), layer 2 (2-3 km), and layer 3 (3-5 km) (Fig. 10a). The high extinction coefficient in layer 1 was attributed to aerosols remaining in the residual layer from the preceding day. The aerosols in layers 2 and 3 were identified as transported pollution aerosol and dust from China, respectively, because the backward trajectories (Fig. 11) indicated that the layer 2 aerosol had been transported from an urban region, and the layer 3 aerosol had been transported from a desert region. These backward trajectories were calculated with our trajectory model, which was developed following Katsumoto et al. (2002). The temporal and spatial distributions of three-dimensional winds were linearly interpolated from the U. S. National Centers for Environmental Prediction 6-hourly reanalysis data set (Kalnay et al. 1996). The time integration was conducted by the fourth-order Runge-Kutta method. The desert and urban regions shown in Fig. 11 were determined by using data from the Land Cover Type Climate Modeling Grid product (LP DAAC, 2013).

The aerosols in layer 3 had a large asymmetry factor value of more than 0.7 (Fig. 10c), and the coarse mode of the size distribution was dominant (Fig. 10d). Therefore, we interpreted the aerosol in layer 3 as pure dust. The asymmetry factor of the aerosols in layer 2 was about 0.65 or higher (Fig. 10c), and both fine and coarse modes of the size distribution had large values (Fig. 10d). Thus, we interpreted the aerosols in layer 2 as a mixture of the dust and pollution aerosols. The single-scattering albedo was largest in layer 2, and it was mostly constant at about 0.97 above the boundary layer (Fig. 10b); smaller values of the single-scattering albedo, ranging from 0.7 to 0.95, were observed in the boundary layer. Uchiyama et al. (2014) reported that the single-scattering albedo near the surface at Tsukuba is typically from 0.7 to 0.95, based on scattering and absorption coefficients measured by nephelometer and particle soot/absorption photometer. Thus, our estimates of the single-scattering albedo range in the boundary layer were typical.

The solar heating rate calculated from the estimated aerosol optical properties is displayed in Fig. 10e. Although the extinction coefficient was largest in layer 2, the solar heating rate was largest in layer 3. To investigate how the vertical profiles of the single-scattering albedo and asymmetry factor influenced the solar heating rate, we calculated the solar heating rate from the vertical means of the single-scattering albedo and asymmetry factor and then calculated the difference between the solar heating rate calculated from estimated optical properties and that calculated from the vertical means. The difference in the case of the solar heating rate calculated from the vertical mean of the asymmetry factor showed that the solar heating rate decreased in layer 3 and increased in layer 2 (Fig. 10f). Theoretically, a large asymmetry factor increases the downward solar flux and the solar heating rate. Because the asymmetry factor of the pure dust in layer 3 was very large, the solar heating rate was large throughout layer 3 (Fig. 10e), whereas the impact of the single-scattering albedo on the



vertical profile of the solar heating rate was small (not shown). Therefore, the vertical profile of the asymmetry factor played an important role in creating vertical variation in the solar heating rate.

We next estimated the aerosol optical properties and the solar heating rate in the case of transported smoke in 8 May, 2013 (Fig. 12). A high aerosol load was observed at altitudes from 4 to 6 km (Fig. 12a). In this layer, the fine mode dominated the

size distribution (Fig. 12d), and the asymmetry factor was about 0.62 (Fig. 12c). The backward trajectory (Fig. 13) indicated that this aerosol had been transported from the region southeast of Lake Baikal, where forest fires had been observed in early May, 2013 (Fig. 13). We interpreted the high volume of fine mode particles as transported smoke from the forest fires. The single-scattering albedo of the transported smoke was estimated to be about 0.96, which is larger than the typical value for smoke (0.84 to 0.94; Dubovik et al., 2002). This overestimation may be attributed to the inability of our algorithm to

reproduce the vertical profiles of the single-scattering albedo when the fine mode is dominant, as shown by our sensitivity tests (Sect. 3). In the boundary layer, the single-scattering albedo was low and the asymmetry factor was high (Fig. 12b and c). These values may reflect locally emitted soil particles; the land surface at Tsukuba was dry during 10 days before 8 May 2013 (total rain-fall was only 0.5 mm), so in many agriculture and urban development areas the ground was bare.

The solar heating rate (Fig. 12e) was consistent with the vertical profile of the extinction coefficients. The difference

between the solar heating rate calculated using estimated optical properties and that calculated using the vertical mean of the asymmetry factor (Fig. 12e) showed that the solar heating rate increased in the smoke layer, because the asymmetry factor of the smoke aerosol was smaller than the vertical mean value (Fig. 12c). The influence of the single-scattering albedo on the vertical profiles of the solar heating rate was small (not shown).

The SKYLIDAR algorithm showed the detailed vertical structures for the transported dust and smoke and the relationship of

the aerosol vertical profiles to the solar heating rate. Our results suggest that the vertical variation of the asymmetry factor plays an important role in creating vertical variation in the solar heating rate.

### 4.2    Comparisons of columnar properties and surface solar irradiance

To validate the estimated vertical profiles of the aerosol optical properties, direct measurements by air plane or balloon are necessary, but such measurements are not obtained easily. Therefore, we compared the columnar optical properties of the

SKYLIDAR retrievals with SKYRAD.PACK retrievals, which have been evaluated by Che et al. (2008) and Estellés et al. (2012). Because the direct observation of the solar heating rate is also difficult, we compared the surface solar irradiance calculated from the SKYLIDAR retrievals with that measured by pyranometer.

We compared the aerosol optical thickness, the single-scattering albedo, the asymmetry factor, and the normalized volume size distribution in the column for 2012-2013 at Tsukuba between SKYLIDAR and SKYRAD.PACK (Fig. 14). The aerosol

optical thickness at 532 and 1064 nm in the SKYLIDAR retrievals agreed well overall with those of the SKYRAD.PACK





retrievals. Although slightly underestimated, the SKYLIDAR single-scattering albedo at 532 nm agreed well with the SKYRAD.PACK retrieval for aerosol optical thickness of more than 0.2. The single-scattering albedo estimated from the SKYR measurements by SKYRAD.PACK, however, is larger than that of the AERONET retrievals (Che et al. 2008). Thus, the SKYLIDAR results may be close to the AERONET retrievals. Similarly, the asymmetry factor also agreed with those

estimated by SKYRAD.PACK for aerosol optical thickness of more than 0.2. Comparison of the two-year mean of the normalized volume size distribution showed agreement with respect to the fine mode but not the coarse mode; the difference win the coarse mode was due to the assumption of a bi-modal size distribution by our algorithm. In the SKYRAD.PACK retrieval, a second coarse mode was observed at a radius of 10 μm. This second coarse mode was reported by Che et al. (2008) and Estellés et al. (2012), who indicated that it is not observed in AERONET retrievals; therefore, it may be

attributed to the difference in the retrieval algorithm. The assumption of a bi-modal size distribution is not ideal, but it is a simple way to prevent the occurrence of an unrealistic second coarse mode.

Surface solar irradiance calculated from the SKYLIDAR retrievals, and also that calculated from the SKYRAD.PACK retrievals, was compared with that measured by the pyranometer during the two years (Fig. 15). The surface irradiances calculated from both the SKYLIDAR and SKYRAD.PACK retrievals agreed very well with the measurements, with a very

small error of around 10 Wm$^{-2}$ (about 1.7 %). The SKYLIDAR result was slightly better than SKYRAD.PACK result. The remaining part of the error is attributed to the error in the input water vapor content and the assumed optical properties at wavelengths less than 532 nm and greater than 1064 nm (described in Sect. 2.2). In this study, we used the water vapor content observed by a sonde launched at 09:00 JST near our observation site.

The estimated vertical profiles of the aerosol optical properties and the solar heating rate were not validated against direct

measurements in this study, but the columnar properties of the SKYLIDAR retrievals agreed well with the SKYRAD.PACK retrievals, and the surface solar irradiance calculated from the SKYLIDAR retrievals were sufficiently accurate to explain the measured surface solar irradiance.

## 5    Summary

We developed the SKYLIDAR algorithm for estimating the vertical profiles of aerosol optical properties from the sun

photometer SKYR and the lidar MIEL (and HSRL) measurements. The algorithm consists of two retrieval steps. The columnar properties are first estimated from the SKYR measurements and the vertically mean depolarization ratio obtained from the MIEL measurements. Then, the vertical profiles are estimated from the MIEL (and HSRL) measurements and the columnar properties determined in the first step. The finally derived parameters are the vertical profiles of the size distribution, refractive index, extinction coefficient, single-scattering albedo, and asymmetry factor. In addition, we

estimated the vertical profile of the solar heating rate from the SKYLIDAR retrievals.



To evaluate the performance of the algorithm, we conducted the sensitivity tests using simulated SKYR and MIEL (and HSRL) data of the vertical profiles of three different aerosols. The vertical profiles of the size distribution, the extinction coefficient, and the asymmetry factor were well estimated in all the tests. The refractive index and the single-scattering albedo in the case of dust were well estimated, but not in the cases of pollution aerosol, which was characterized by a size distribution with a dominant fine mode and strong light absorption.

We then applied the SKYLIDAR algorithm to actual SKYR and MIEL measurements obtained in 2012 and 2013 in Tsukuba, Japan. Our algorithm showed the detailed vertical structures for transported dust and smoke. In addition, the vertical profiles of the solar heating rate were estimated from the SKYLIDAR retrievals, and the relationship of the aerosol optical properties to the solar heating rate was investigated. The results suggest that the vertical profile of the asymmetry factor plays an important role in creating vertical variations of the solar heating rate.

To evaluate the validity of the SKYLIDAR retrievals and the solar heating rate, we compared the columnar properties between SKYLIDAR and SKYRAD.PACK retrievals during 2012 and 2013, and we compared the surface solar irradiances calculated from the SKYLIDAR retrievals with those measured by pyranometer. The columnar properties of the SKYLIDAR retrievals agreed well with those of the SKYRAD.PACK retrievals when the aerosol optical thickness at 532 nm was more than 0.2. The calculated surface solar irradiances also agreed well with the pyranometer measurements; the mean error was only 1.7 %, despite the assumption that extended the aerosol optical properties at 532 and 1064 nm to the broadband wavelength regions. The columnar properties of the SKYLIDAR retrievals agreed with those estimated by the widely used method of SKYRAD.PACK, and the SKYLIDAR retrievals were sufficiently accurate to evaluate surface solar irradiance.

In this study, we focused on the SKYR (SKYNET) and MIEL (AD-Net) measurements, but the SKYLIDAR algorithm can be applied to an another data set similar to SKR and MIEL measurements. This flexibility is expected to be useful for investigating the temporal and spatial distribution of aerosols at different observational sites. In addition, the minimization procedure with our developed logarithm transformations, which works well for hundreds of estimated parameters, is useful for the various remote sensing.

**Acknowledgements**

This work was supported by the Japan Society for the Promotion of Science KAKENHI Grant No. 24510026. The authors are grateful to the OpenCLASTR project for allowing us to use the SKYRAD.PACK (sky radiometer analysis package) in this research. NCEP reanalysis data were provided by the NOAA/OAR/ESRL PSD, Boulder, Colorado, USA, from its Web site at http://www.esrl.noaa.gov/psd/. The MODIS MCD12C1 product was retrieved from the online Data Pool, courtesy of



the NASA EOSDIS Land Processes Distributed Active Archive Center (LP DAAC), USGS/Earth Resources Observation and Science (EROS) Center, Sioux Falls, South Dakota, https://lpdaac.usgs.gov/data_access/data_pool.

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



Table 1. Microphysical and optical properties and vertical profiles of the aerosols used in the sensitivity tests.

| Aerosol | Components | Size distribution | | Refractive index at 500 nm | | Relative weight in total optical thickness at 500 nm | Vertical profile |
| | | Mode Radius (µm) | Mode Width | Real | Imaginary | | |
|---|---|---|---|---|---|---|---|
| Continental average | Water-soluble | 0.18 | 0.81 | 1.44 | 0.0026 | 0.90 | $\exp(-z/H)$, H = 8 km |
| | Soot | 0.05 | 0.69 | 1.75 | 0.45 | 0.07 | $\exp(-z/H)$, H = 4 km |
| | Insoluble | 5.98 | 0.92 | 1.53 | 0.008 | 0.03 | $\exp(-z/H)$, H = 2 km |
| Transported dust | Dust | 3.23 | 0.79 | 1.53 | 0.0078 | 0.25 | $\frac{1}{\sqrt{2\pi}\sigma}\exp\left(-\frac{(z-z_c)}{2\sigma^2}\right)$, $z_c$ = 3.5 km, $\sigma$ = 0.4 km |
| | Water-soluble | 0.18 | 0.81 | 1.44 | 0.0026 | 0.67 | $\exp(-z/H)$, H = 8 km |
| | Soot | 0.05 | 0.69 | 1.75 | 0.45 | 0.05 | $\exp(-z/H)$, H = 4 km |
| | Insoluble | 5.98 | 0.92 | 1.53 | 0.008 | 0.03 | $\exp(-z/H)$, H = 2 km |
| Transported pollution | Water-soluble | 0.18 | 0.81 | 1.44 | 0.0026 | 0.08 | $\frac{1}{\sqrt{2\pi}\sigma}\exp\left(-\frac{(z-z_c)}{2\sigma^2}\right)$, $z_c$ = 3.5 km, $\sigma$ = 0.4 km |
| | Soot | 0.05 | 0.69 | 1.75 | 0.45 | 0.03 | $\frac{1}{\sqrt{2\pi}\sigma}\exp\left(-\frac{(z-z_c)}{2\sigma^2}\right)$, $z_c$ = 3.5 km, $\sigma$ = 0.4 km |
| | Water-soluble | 0.18 | 0.81 | 1.44 | 0.0026 | 0.79 | $\exp(-z/H)$, H = 8 km |
| | Soot | 0.05 | 0.69 | 1.75 | 0.45 | 0.06 | $\exp(-z/H)$, H = 4 km |
| | Insoluble | 5.98 | 0.92 | 1.53 | 0.008 | 0.03 | $\exp(-z/H)$, H = 2 km |





## (a) Step 1: Columnar property

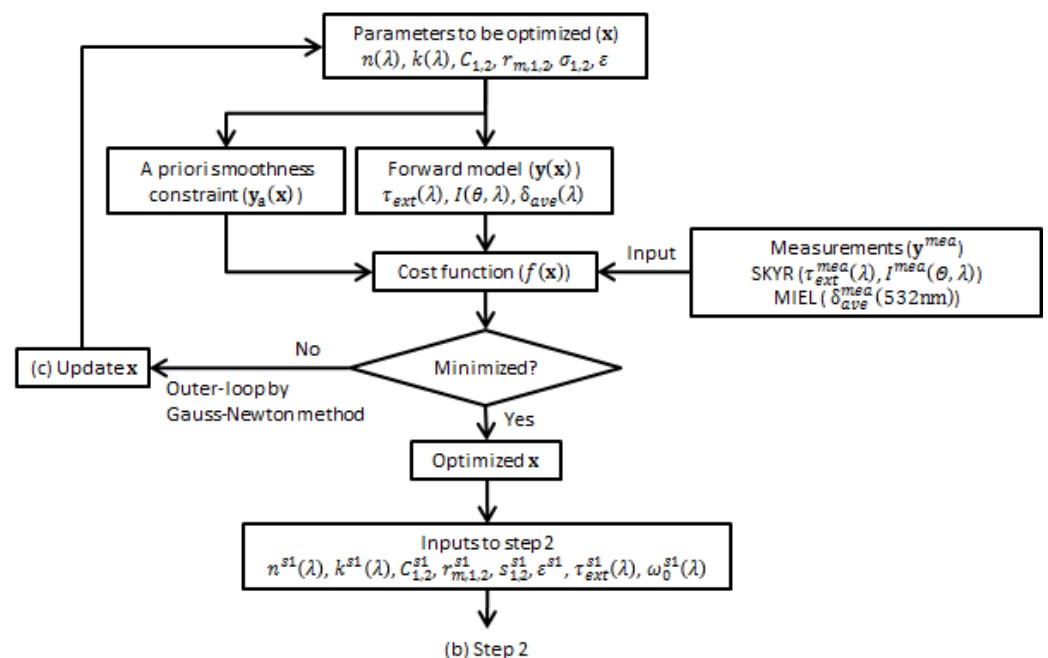

## (b) Step 2: Vertical profile

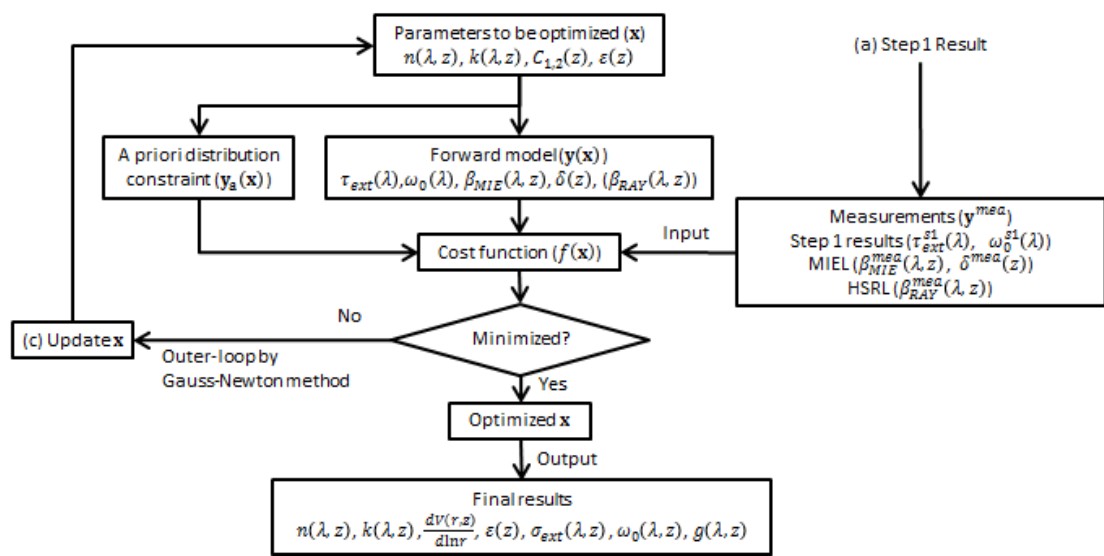



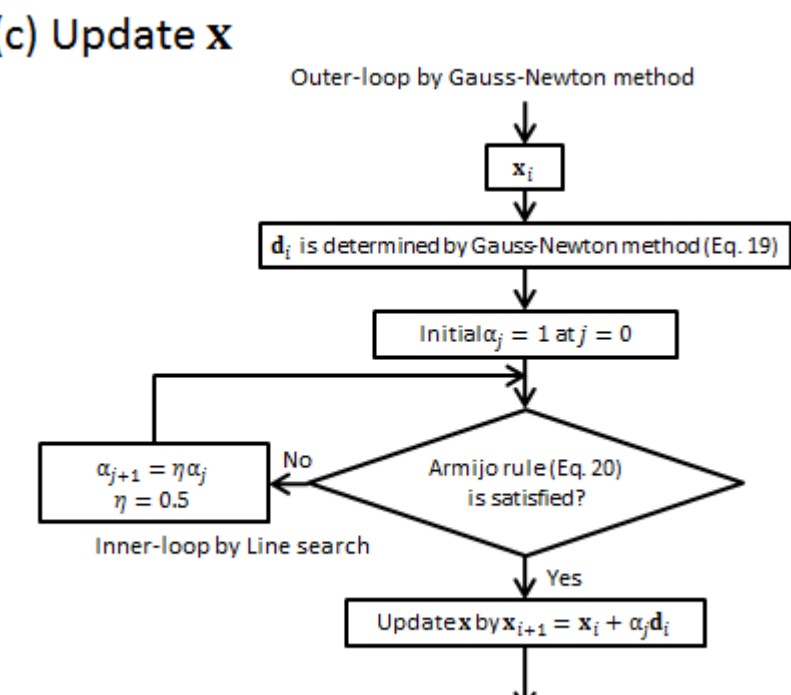

Figure 1. Schematic diagrams of the retrieval procedures: (a) step 1; (b) step 2; and (c) updating parameters **x**.



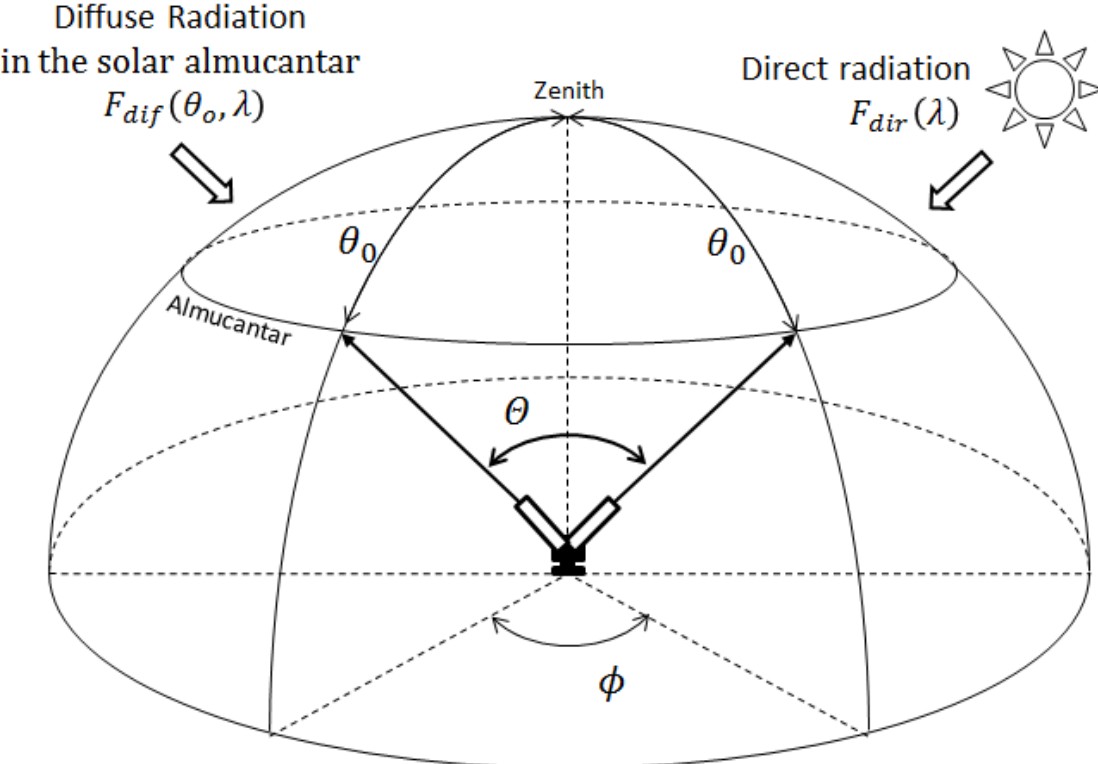

Figure 2. Observation of the sky radiometer in the solar almucantar geometry.



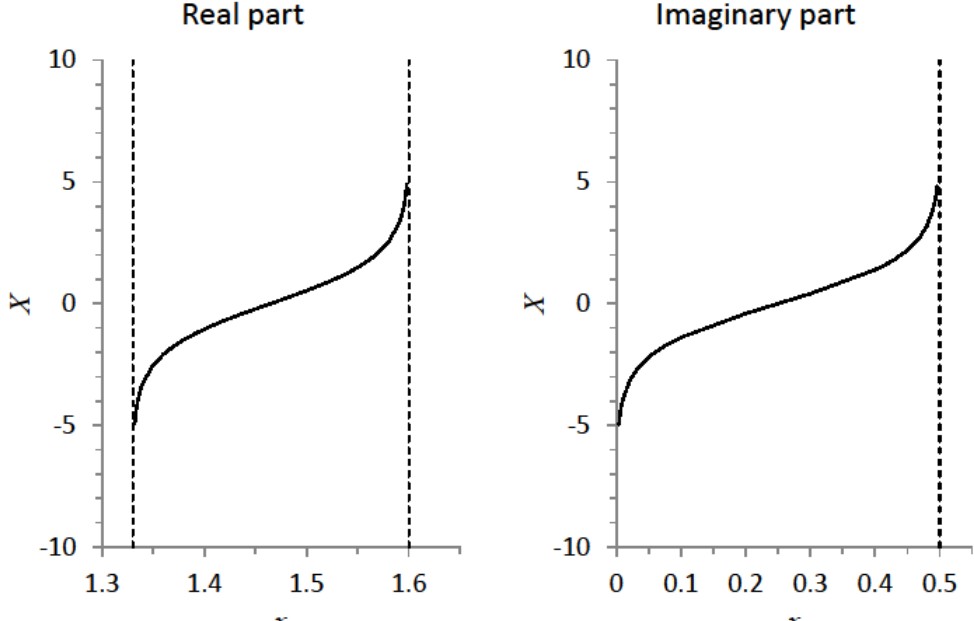

Figure 3. Transformation of the real and imaginary parts of the refractive indices by Eq. (24). Minimum and maximum values of the real part are 1.33 and 1.6, and those of the imaginary part are 0.0005 and 0.5.



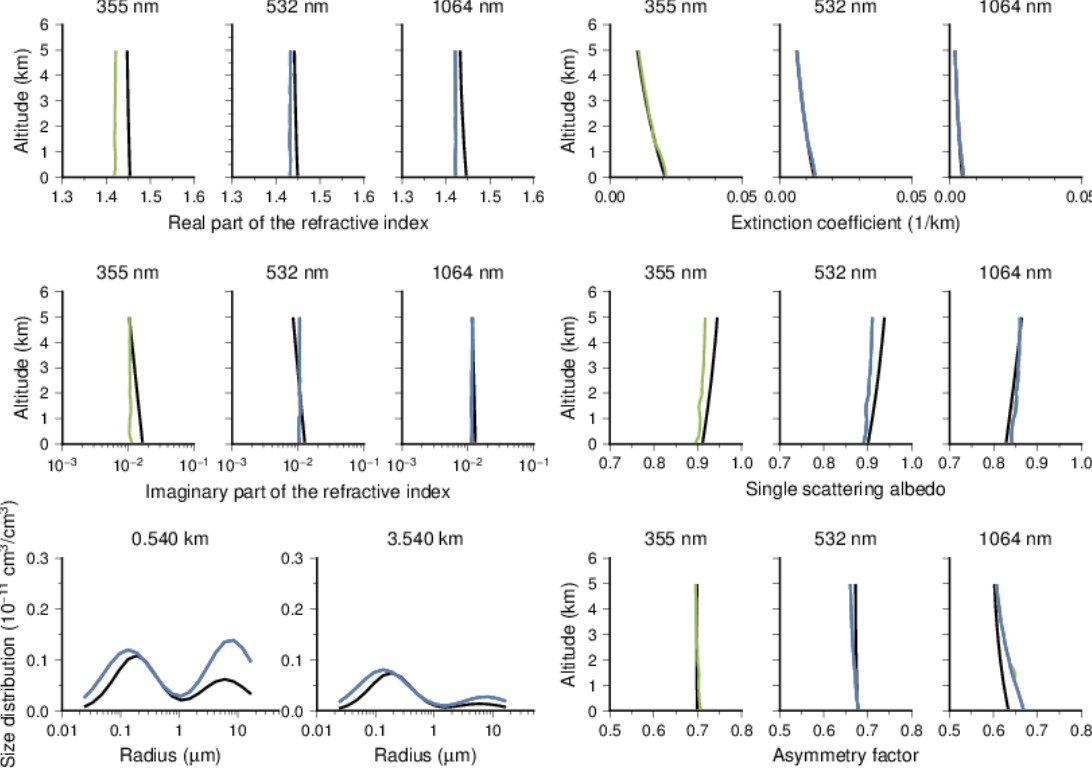

Figure 4. Retrieval results from the simulated data for the continental average aerosol: the real and imaginary parts of the refractive index, the size distribution, the extinction coefficients, the single-scattering albedo, and the asymmetry factor. "True" values are shown by the black lines, and the colored lines indicate retrievals from Type 1 data set (MIEL data at 532 and 1064 nm) (blue); Type 2 data set (MIEL data at 532 and 1064 nm, and HSRL data at 532 nm) (red); and Type 3 data set (MIEL data at 532 and 1064 nm, and HSRL data at 355 and 532 nm) (green). The red and blue lines completely overlap.




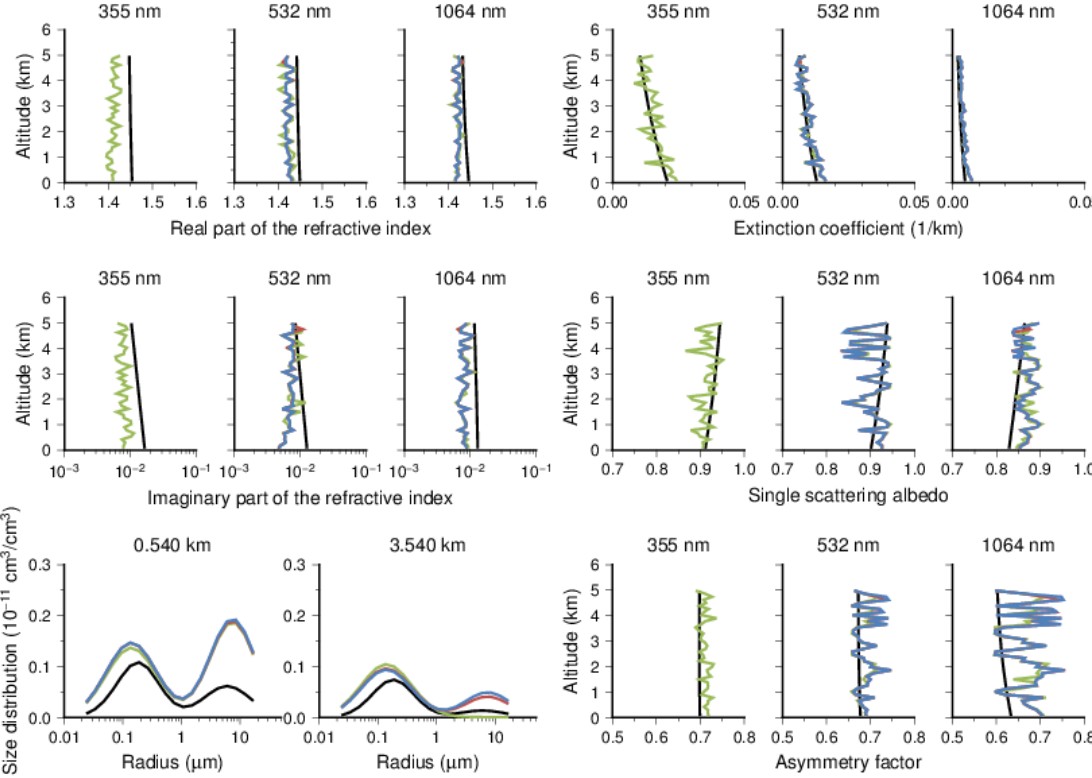

Figure 5. Same as Fig. 4, but showing the retrieval results from the simulated data with random errors for the continental average aerosol.




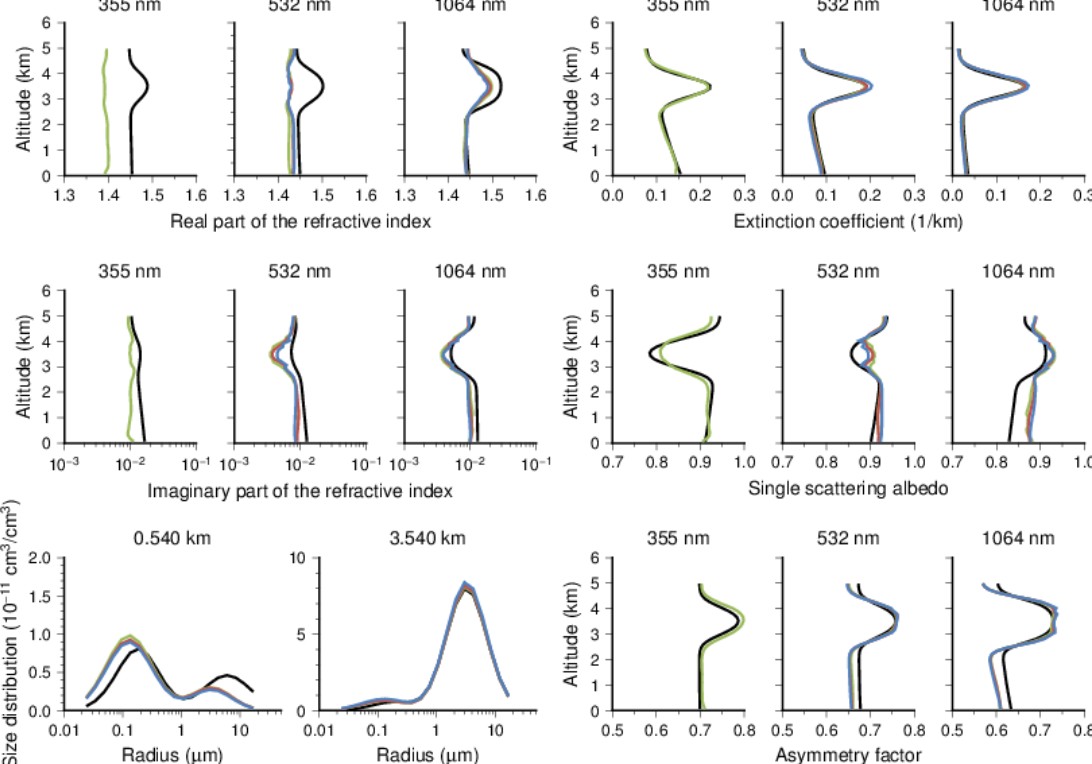

Figure 6. Retrieval results from the simulated data for the transported dust: the real and imaginary parts of the refractive index, the size distribution, the extinction coefficients, the single-scattering albedo, and the asymmetry factor. "True" values are shown by the black lines, the colored lines indicate retrievals from Type 1 data set (MIEL data at 532 and 1064 nm) (blue); Type 2 data set (MIEL data at 532 and 1064 nm, and HSRL data at 532 nm) (red); and Type 3 data set (MIEL data at 532 and 1064 nm, and HSRL data at 355 and 532 nm) (green).



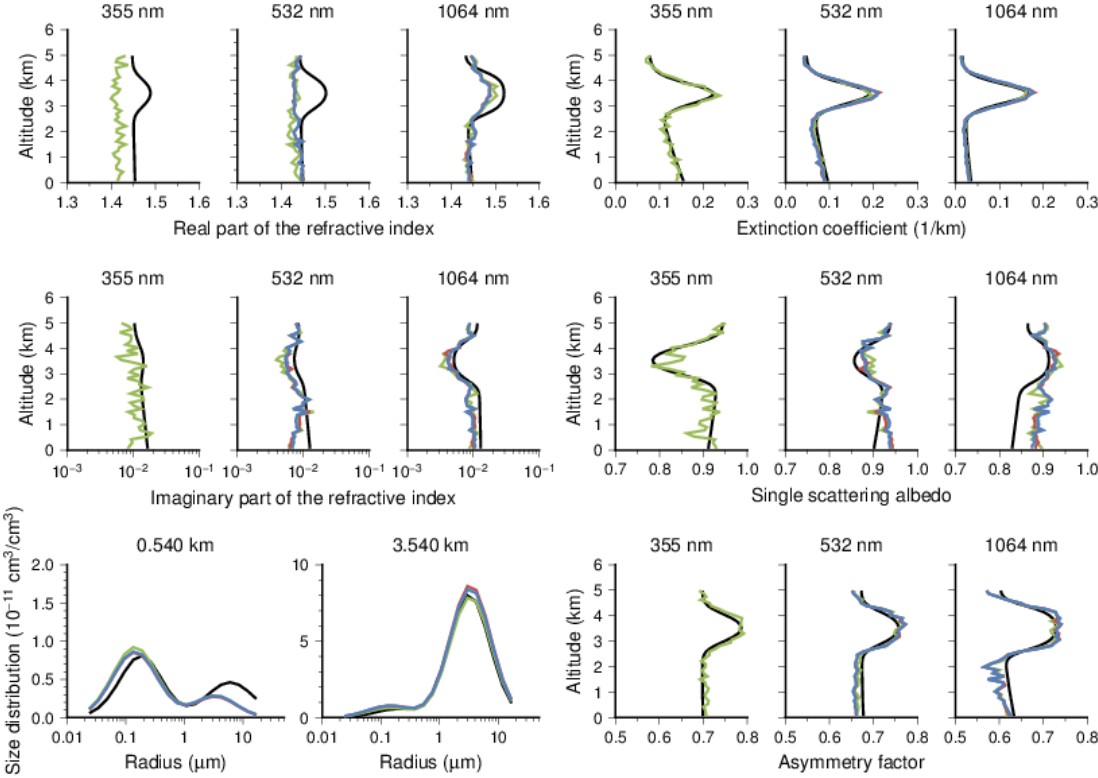

Figure 7. Fig. 6, but showing the retrieval results from the simulated data with random errors for the transported dust.




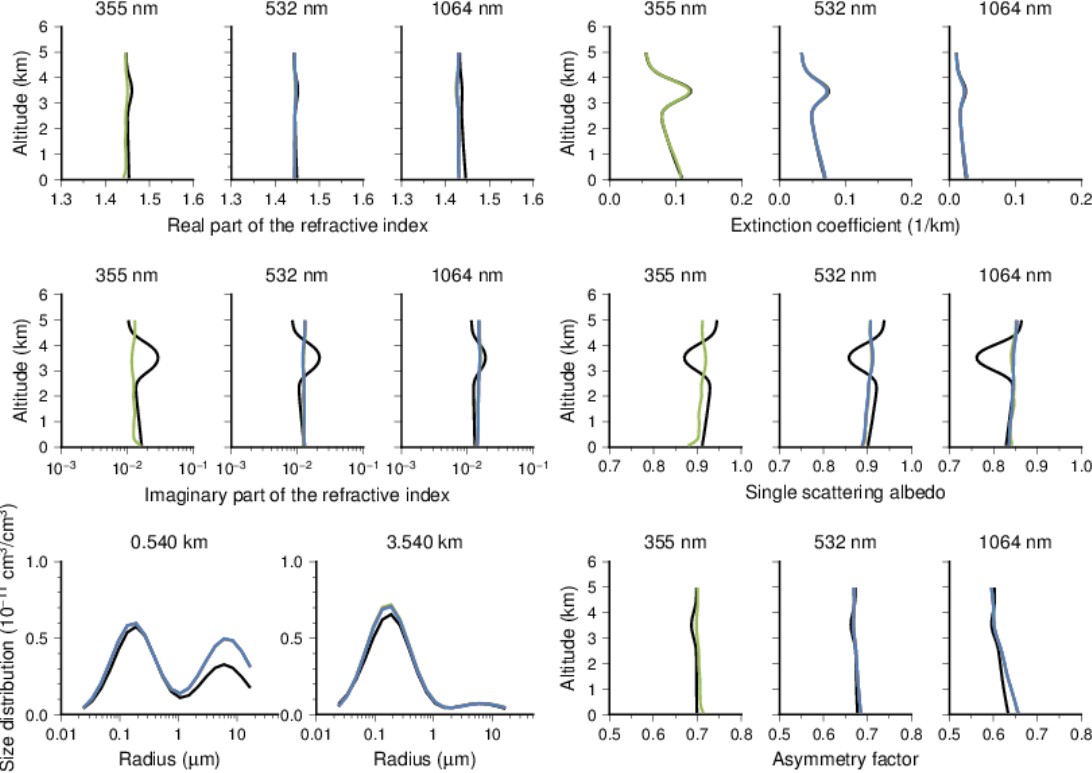

Figure 8. Retrieval results from the simulated data for the transported pollution aerosol: the real and imaginary parts of the refractive index, the size distribution, the extinction coefficients, the single-scattering albedo, and the asymmetry factor. "True" values are shown by the black lines, the colored lines indicate retrievals from Type 1 data set (MIEL data at 532 and 1064 nm) (blue); Type 2 data set (MIEL data at 532 and 1064 nm, and HSRL data at 532 nm) (red); and Type 3 data set (MIEL data at 532 and 1064 nm, and HSRL data at 355 and 532 nm) (green). The red and blue lines completely overlap.





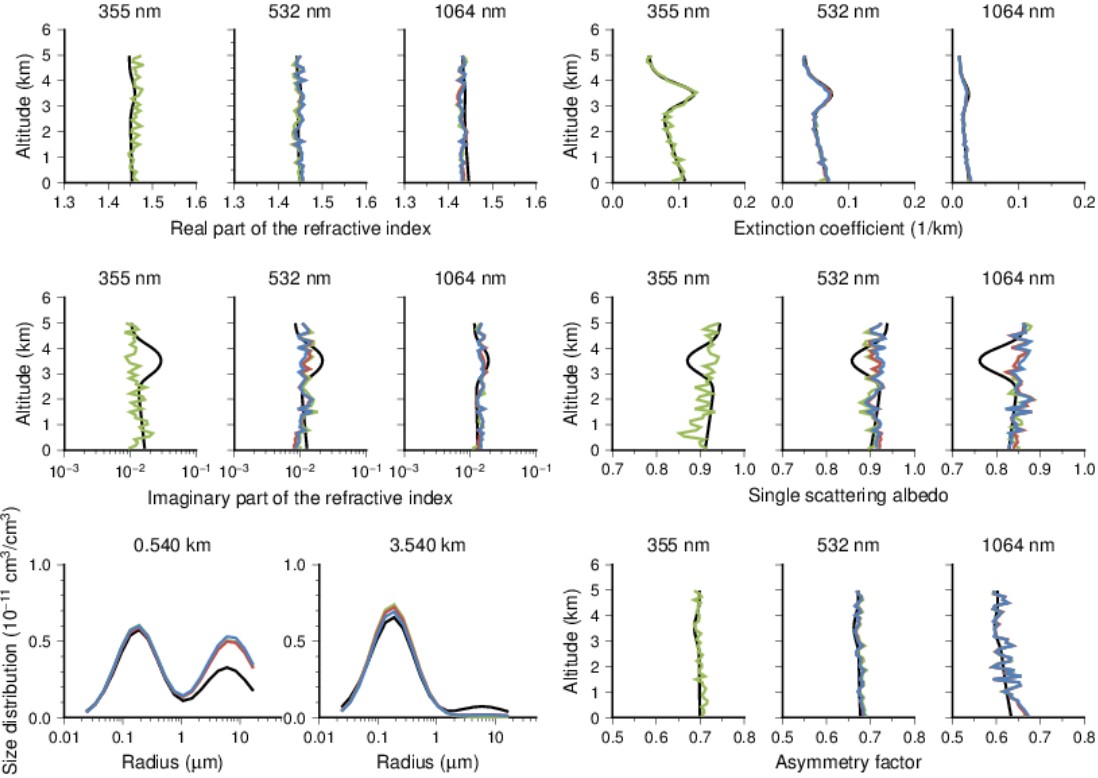

Figure 9. Same as Fig. 7, but showing the retrieval results from the simulated data with random errors for the transported pollution aerosol.



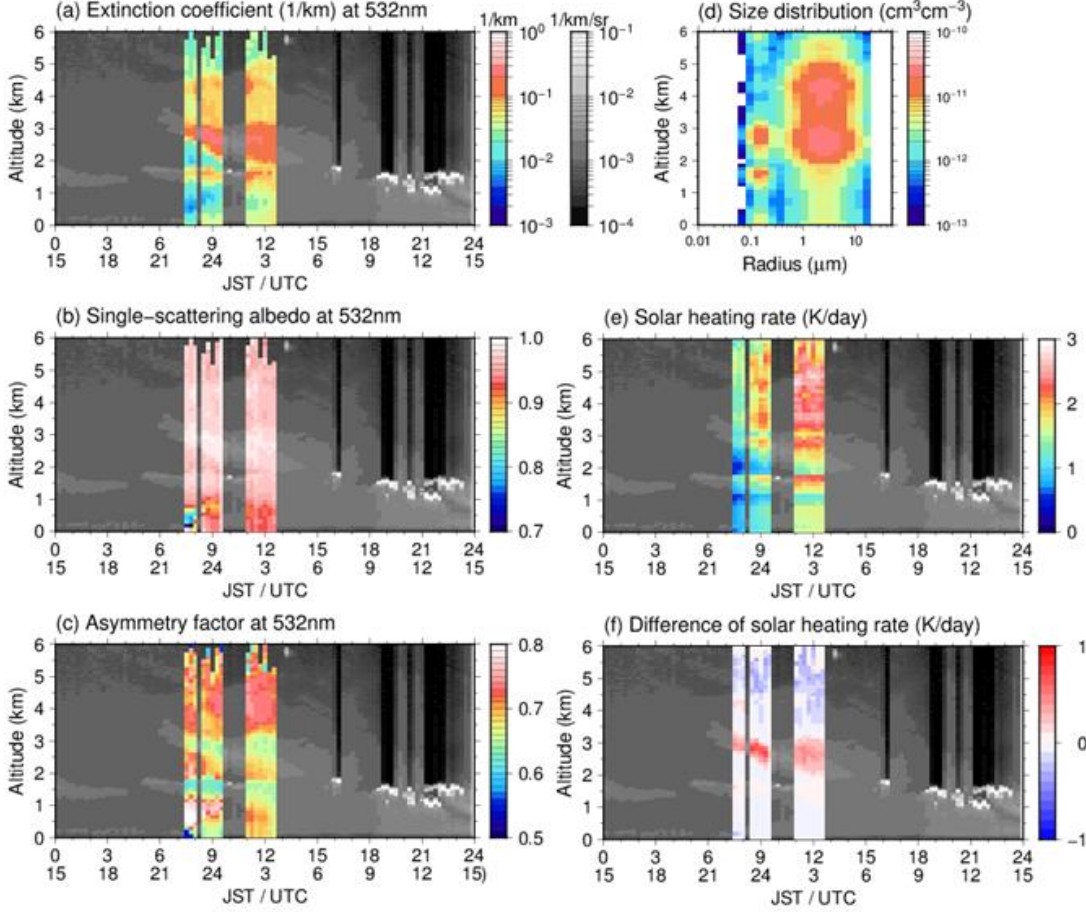

Figure 10. Retrieval results for the transported dust in 2 April 2012 in Tsukuba, Japan: (a) extinction coefficient at 532 nm, (b) single-scattering albedo at 532 nm, (c) asymmetry factor at 532 nm, (d) size distribution, (e) solar heating rate, and (f) solar heating rate difference from that calculated using the vertical mean of the asymmetry factor. The gray shading shows the attenuated backscatter coefficient at 532 nm by MIEL.





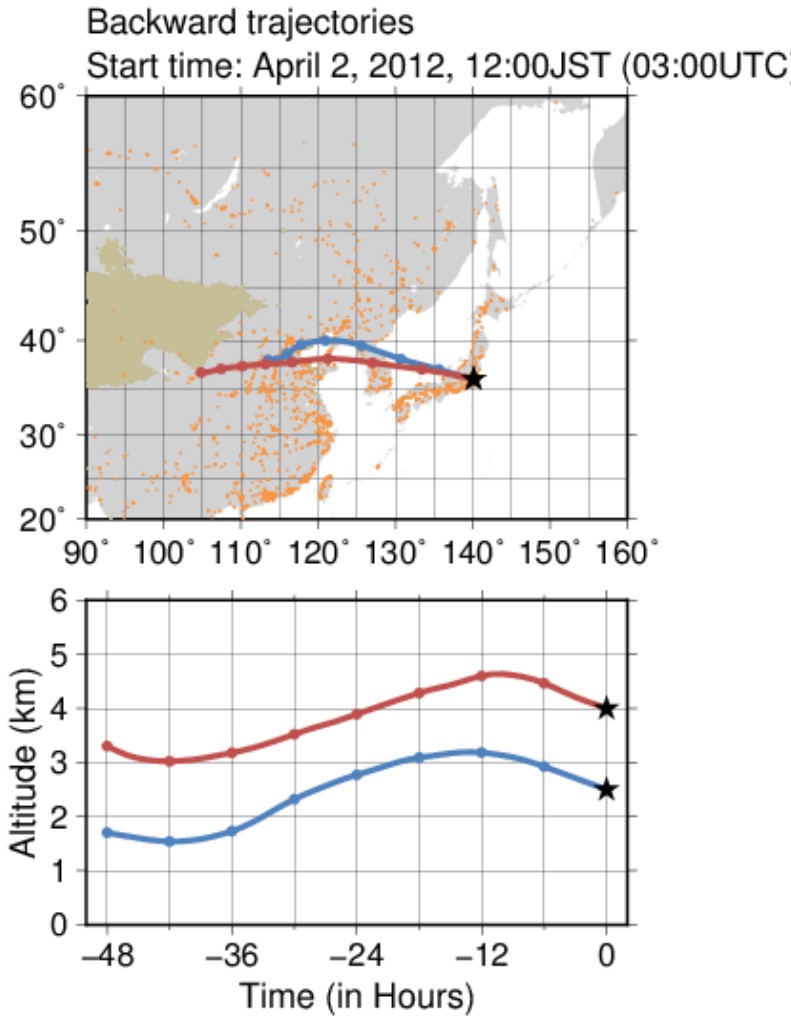

Figure 11. Two-days backward trajectories of air masses observed on 2 April 2012. The ochre region and orange areas in the upper panel indicate desert region and urban regions, respectively.



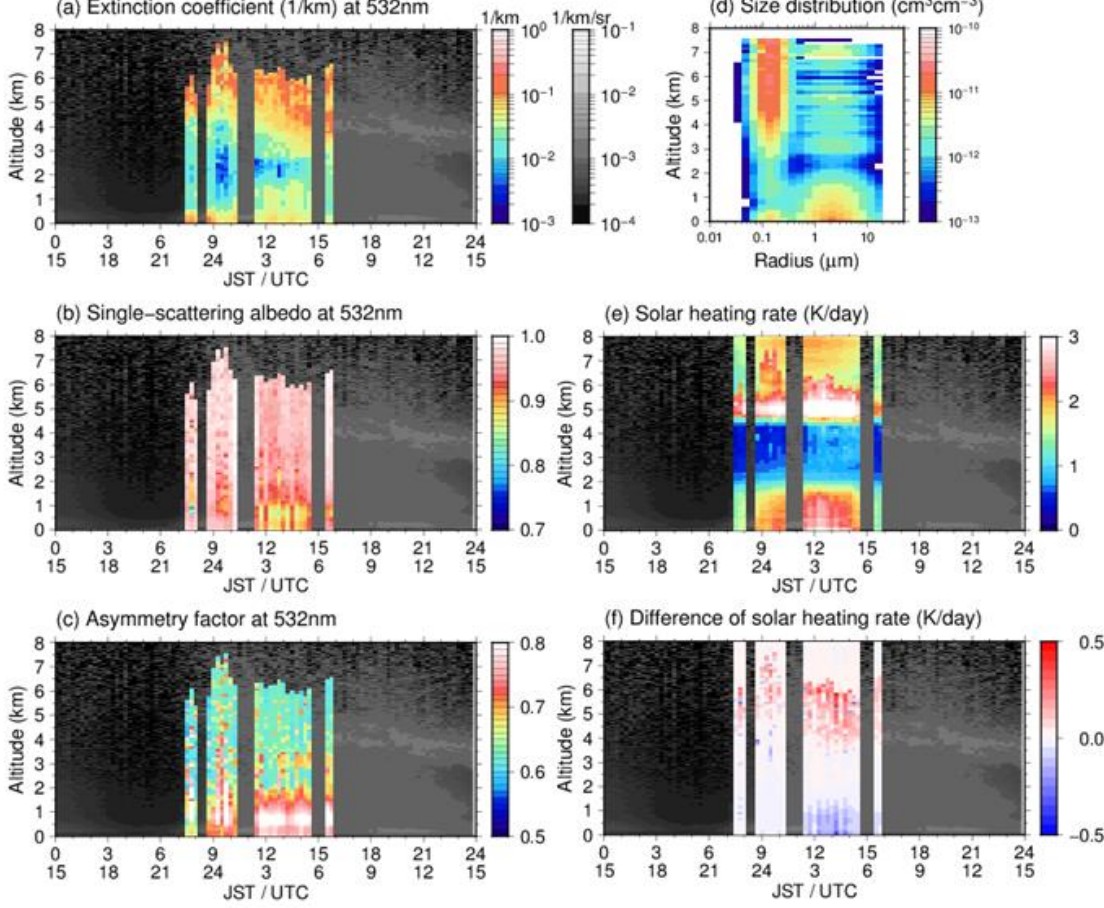

Figure 12. Retrieval results for the transported smoke in 8 May 2013 in Tsukuba, Japan: (a) extinction coefficient at 532 nm, (b) single-scattering albedo at 532 nm, (c) asymmetry factor at 532 nm, (d) size distribution, (e) solar heating rate, and (f) solar heating rate difference from that calculated using the vertical mean of the asymmetry factor. The gray shading is the attenuated backscatter coefficient at 532 nm by MIEL.





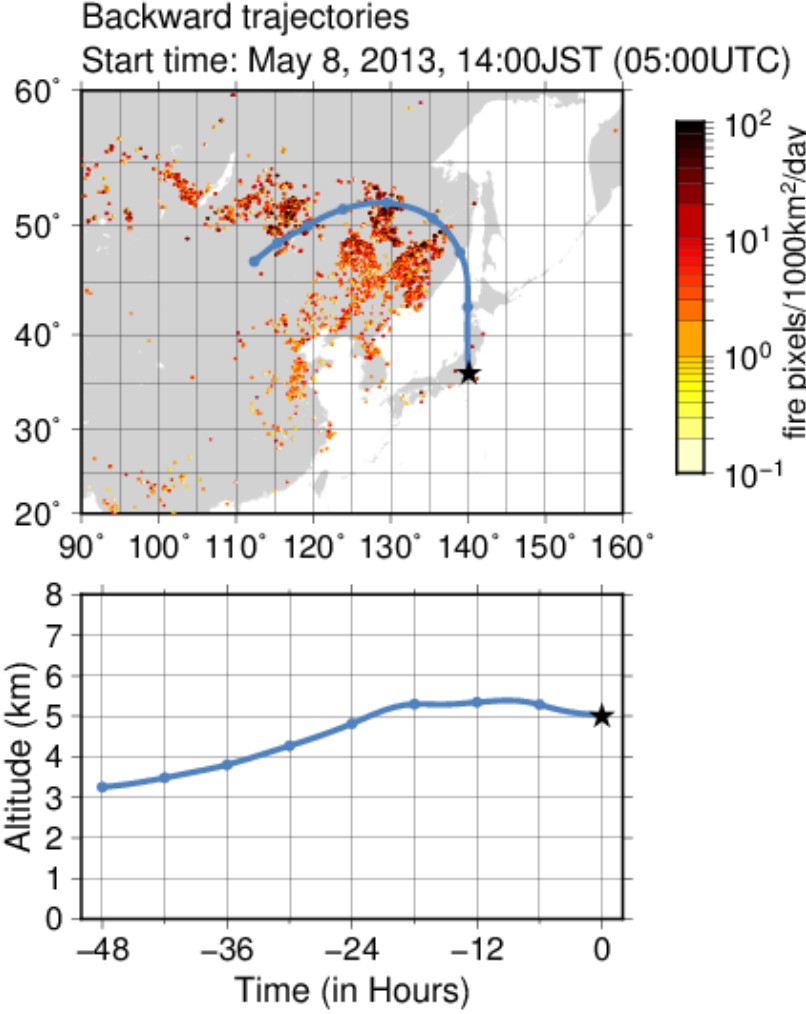

Figure 13. Two-days backward trajectory of air mass observed on 8 May 2013. The upper panel also shows fire activity from 1 to 9 May 2013 (color scale), based on data of MODIS active-fire products (NASA Earth Observations http://neo.sci.gsfc.nasa.gov/).





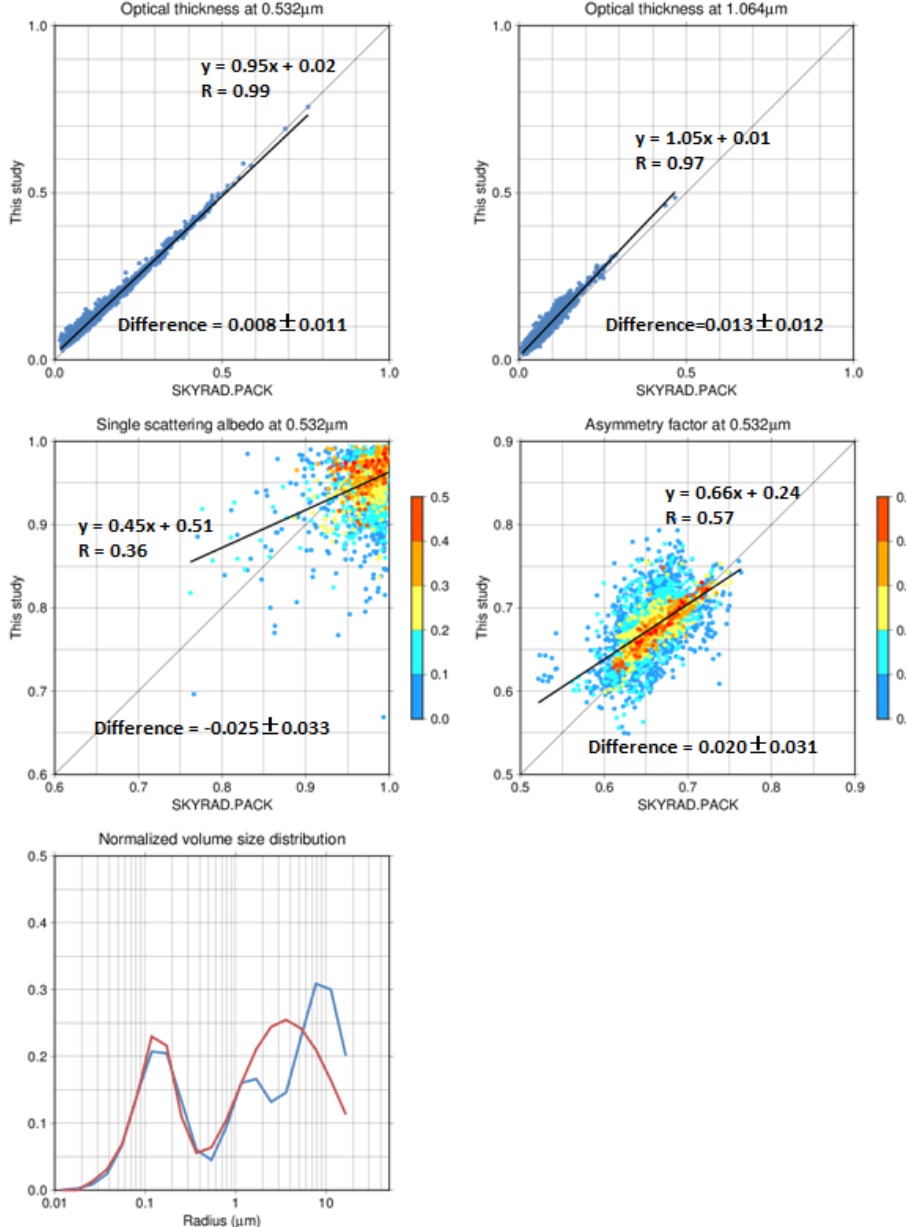

Figure 14. Comparisons of the columnar optical properties estimated by SKYLIDAR algorithm (this study) and SKYRAD.PACK during 2012 and 2013 in Tsukuba, Japan. The colors in the single-scattering albedo and asymmetry factor panels show the aerosol optical thickness at 532 nm. In normalized volume size distribution panel, red indicates the SKYLIDAR result, and blue indicates the SKYRAD.PACK result.





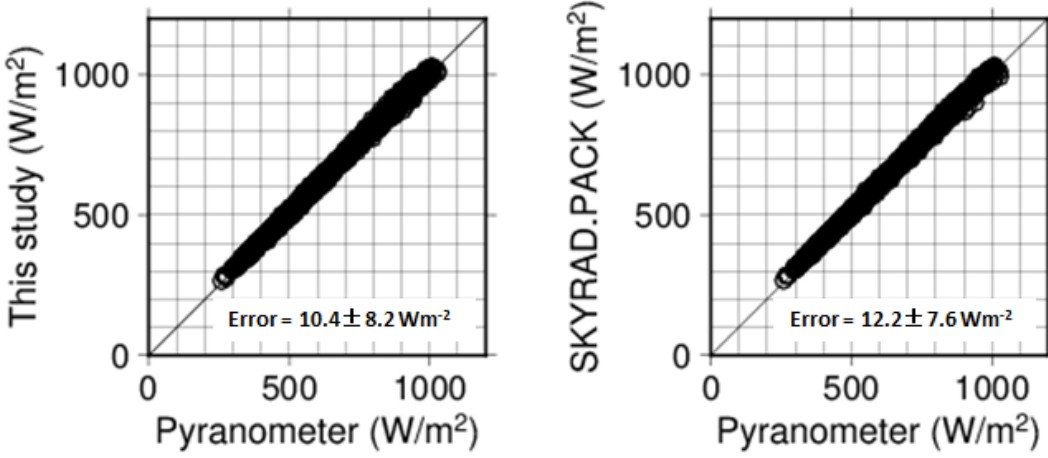

Figure 15. Comparisons of the surface solar irradiance calculated from the SKYLIDAR (left) and SKYRAD.PACK (right) retrievals with by the pyranometer measurements during 2012 and 2013 in Tsukuba, Japan.

