# Peer review of "Vertical profiles of aerosol optical properties and the solar heating rate estimated by combining sky radiometer and lidar measurements"

_Atmospheric Measurement Techniques, 2016_

## Referee Comment (RC1) · Anonymous Referee #1 · 22 Apr 2016

The paper describes a retrieval scheme where passive and active ground-based measurements are combined with the aim of retrieving vertical profiles of aerosol optical properties, which are then used to infer solar heating rate profiles. The algorithm has been tested on a number of synthetic scenarios – displaying a good capability to retrieve most of the considered aerosol properties when the aerosol load is not too small – and then applied to real measurements. AOT and SSA and asymmetry factors retrieved on real measurements show good agreement with those retrieved using an existing algorithm, and direct comparisons between retrieved surface solar irradiances and corresponding pyranometer measurements also show good agreement. I think the paper is generally well written and fits the scope of AMT. I do not have major criticism

towards the presented work, but I would just like to suggest a number of minor revisions the Authors may want to take into consideration before the manuscript is published.

1. Since I did not know about SKYRAD.PACK before reading this paper, I had to look it up in order to understand what you mean. I think this is not a good feature in an abstract, so you may want to spend at least a couple of words to explain what SKYRAD.PACK is. For example, you may say something like "We then compared the optical properties retrieved with the SKYLIDAR algorithm to those produced with the more established scheme SKYRAD.PACK".

2. P2, L15. By Mie lidar you mean the basic, conventional lidar, right?

3. P4, L6. "langley method" -> "Langley method".

4. P4, L15. It may be worthwhile to remind the reader that $\cos^2(\theta_0)$ and $\sin^2(\theta_0)$ in the expression of the scattering angle appear because we are in the almucantar plane, where $\theta = \theta_0$.

5. P5, L26. I do not agree on the fact that your algorithm is a maximum likelihood method, because you have a priori constraints and the pdf you maximize is an a posteriori one. Therefore it would be more proper to say that your algorithm is a maximum a posteriori (MAP) scheme. I am aware that other papers about aerosol retrievals claim that the method you present is maximum likelihood, but if one wants to keep consistency with standard statistical terminology (which I think would be a good thing) I am almost certain that it is not. To the best of my knowledge, maximum likelihood methods seek the maximum of $P(\mathbf{y}^{mea}|\mathbf{y}(\mathbf{x}))$, which is different from $P(\mathbf{y}(\mathbf{x})|\mathbf{y}^{mea})$ (Lehmann and Casella, 1998; Robert, 2007), and usually need a sample of independent and identically distributed realizations of the observation vector $\mathbf{y}^{mea}$, which are rarely available in remote sensing, unless you repeat the measurement of the full vector $\mathbf{y}^{mea}$ multiple times under

identical conditions and use all the repeated measurements to build a joint pdf that you later maximize with respect to $\mathbf{x}$. This remark may seem pedantic, but the difference between MAP and maximum likelihood has some remarkable consequences. For instance, maximum likelihood estimates are invariant to reparameterizations (i.e., if $\mathbf{x}_0$ is the optimal estimate of $\mathbf{x}$ then $g(\mathbf{x}_0)$ is the optimal estimate of $g(\mathbf{x})$ for any bijective function $g$), whereas MAP estimates are not. MAP estimates are numerically equal to maximum likelihood if a flat (improper) prior is used, but of course being numerically equal does not mean that they are the same thing.

6. P6, L6. If $\mathbf{W}$ is the covariance matrix then its diagonal elements are the squares of the standard errors in the measurements (not the errors themselves) and in equation (4) you should replace $\mathbf{W}^2$ with $\mathbf{W}$. If, instead, the diagonal elements of $\mathbf{W}$ are the errors then it is correct to keep $\mathbf{W}^2$ in eq. (4) but you should say that $\mathbf{W}^2$ (not $\mathbf{W}$) is the covariance matrix.

7. P6, L15 and eq. (5) and (6). I do not find the notation very clear. Wouldn't it be better to just say that $\mathbf{y}_{mea}$ contains $\tau_{ext}^{mea}$ at all wavelengths, $I^{mea}$ at all wavelengths and scattering angles and $\delta_{ave}^{mea}$ at 532 nm? Or, if you want to use an equation, you may use the more established notation

$$\mathbf{y}^{mea} = [\tau_{ext}^{mea}(\lambda_1), \ldots \ldots, \tau_{ext}^{mea}(\lambda_n), I^{mea}(\Theta_1, \lambda_1), I^{mea}(\Theta_2, \lambda_1), \ldots$$

$$\ldots, I^{mea}(\Theta_m, \lambda_n), \delta_{ave}^{mea}(532\,\text{nm})]$$

or you can summarize the content of the vector in a table. Similarly, in eq. (6) wouldn't it be clearer if you just state that x contains real and imaginary part of refractive index at all SKYR wavelengths plus all the other parameters?

8. P9, equations. Again, the notation of the formulas looks a bit confusing to me. The suggestions I would give you are similar to those given in the previous point.

9. P11, eq.(21). $\mathbf{W}$ and $\mathbf{W}_a$ should be uppercase. Please re-check whether there should be $\mathbf{W}^2$ or $\mathbf{W}$ depending on the choice you make about the meaning of $\mathbf{W}$ (see point 6).

10. P12, L24. "In actuality" -> "Actually" or "Eventually".

11. P13, L27. It is not clear to me what does "the broadband wavelength regions of solar radiation" mean.

12. P13, L28. "The wavelengths of the optical properties" -> "The wavelengths at which the optical properties . . . are obtained".

13. P14, L2. Which kind of interpolation did you use to compute the refractive index between 532 and 1064 nm? I guess linear is enough for the real part, but what about the imaginary part?

14. P15, L2. "other than" -> "except" (it sounds clearer).

15. P15, L3. Is the size distribution only retrieved at 540 and 3540 m? Is that enough to call that a "profile of the aerosol size distribution"? Furthermore, you assume a bimodal lognormal size distribution and – if I understood correctly – you fit mode radii and widths for the entire column in step 1, whereas in step 2 you keep those values fixed but update $C_1$ and $C_2$ per layer. Then, I guess, your estimate of the aerosol size distribution profile is controlled by these two parameters, right? Wouldn't it be more informative to compare and plot the profiles of $C_1(z)$ and $C_2(z)$?

16. P15, L7. "plotted" -> "shows"; "the aerosol optical thickness of" -> "an aerosol optical thickness of".

17. P15, L8-9. You say the profiles of the refractive index were not estimated well, but doesn't that only apply to the imaginary part? You may want to specify that.

[Figure]

18. P19, L21. SKR -> SKYR ?

REFERENCES

Lehmann, E.L., and Casella, G., "Theory of Point Estimation", II edition, Springer Texts in Statistics, 1998

Robert, C. P., "The Bayesian Choice. From Decision-Theoretic Foundations to Computational Implementation", Springer Texts in Statistics, 2007

---

## Referee Comment (RC2) · Anonymous Referee #1 · 2 May 2016

I just noticed that in my remark N.1 about SKYRAD.PACK I forgot to mention explicitly that it was relative to the abstract. Perhaps it was already clear from the content of my remark, but just to be sure I prefer to say that explicitly.

---

## Author Comment (AC1) · 16 May 2016

Reply to Referee 1 comment

The authors thank the referee for a through reading of the manuscript and helpful comments. We have responded to the all comments, namely.

The changes to meet your comments are marked with yellow or blue background in the revised manuscript.

1. Since I did not know about SKYRAD.PACK before reading this paper, I had to look it up in order to understand what you mean. I think this is not a good feature in an abstract, so you may want to spend at least a couple of words to explain what SKYRAD.PACK is. For example, you may say something like "We then compared the optical properties retrieved with the SKYLIDAR algorithm to those produced with the more established scheme SKYRAD.PACK".

I just noticed that in my remark N.1 about SKYRAD.PACK I forgot to mention explicitly   that it was relative to the abstract. Perhaps it was already clear from the content of my   remark, but just to be sure I prefer to say that explicitly.

Ans.) We have revised the sentence (P1, L22).

2. P2, L15. By Mie lidar you mean the basic, conventional lidar, right?

Ans.) Yes. We have changed "Mie lidar" to "conventional elastic backscatter lidar" (P2, L15).

3. P4, L6. "langley method" -> "Langley method".

Ans.) We have corrected it (P4, L6).

4. P4, L15. It may be worthwhile to remind the reader that $\cos^2(\theta_0)$ and $\sin^2(\theta_0)$ in the expression of the scattering angle appear because we are in the almucantar

plane, where $\theta = \theta_0$.

Ans.) We have modified Eqs (1) and (2), Fig 2, and the sentence after Eq. (2) (P4, L13-18).

5. P5, L26. I do not agree on the fact that your algorithm is a maximum likelihood method, because you have a priori constraints and the pdf you maximize is an a posteriori one. Therefore it would be more proper to say that your algorithm is a maximum a posteriori (MAP) scheme. I am aware that other papers about aerosol retrievals claim that the method you present is maximum likelihood, but if one wants to keep consistency with standard statistical terminology (which I think would be a good thing) I am almost certain that it is not. To the best of my knowledge, maximum likelihood methods seek the maximum of $P(\mathbf{y}^{mea}|\mathbf{y}(\mathbf{x}))$, which is different from $P(\mathbf{y}(\mathbf{x})|\mathbf{y}^{mea})$ (Lehmann and Casella, 1998; Robert, 2007), and usually need a sample of independent and identically distributed realizations of the observation vector $\mathbf{y}^{mea}$, which are rarely available in remote sensing, unless you repeat the measurement of the full vector $\mathbf{y}^{mea}$ multiple times under identical conditions and use all the repeated measurements to build a joint pdf that you later maximize with respect to $\mathbf{x}$. This remark may seem pedantic, but the difference between MAP and maximum likelihood has some remarkable consequences. For instance, maximum likelihood estimates are invariant to reparameterizations (i.e., if $\mathbf{x}_0$ is the optimal estimate of $\mathbf{x}$ then $g(\mathbf{x}_0)$ is the optimal estimate of $g(\mathbf{x})$ for any bijective function $g$), whereas MAP estimates are not. MAP estimates are numerically equal to maximum likelihood if a flat (improper) prior is used, but of course being numerically equal does not mean that they are the same thing.

Ans.) The formulation in the step 2 retrieval is just that of the MAP scheme. We agree to your comments. In the atmospheric remote sensing, it is difficult to obtain the solution without any prior information. The MAP scheme considers the prior information basically, but the method of the maximum likelihood does not. We have revised the formulation as the MAP scheme in the revised manuscript. However, the smoothness constraint in the step 1 retrieval could not be formulated from $(\mathbf{x} - \mathbf{x}_a)^T (\mathbf{W}_a^2)^{-1} (\mathbf{x} - \mathbf{x}_a)$. We introduced the smoothness constraint as a virtual measurement by considering the constraint as being of the same nature as a measurement (Dubovik and King, 2000) (P8, L11-23). These changes are marked

with blue background. The equations in the revised manuscript are changed, but they are numerically equal to previous ones and all the results are not changed.

Dubovik, O., and King, M. D. (2000), A flexible inversion algorithm for retrieval of aerosol optical properties from sun and sky radiance measurements, J. Geophys. Res., 105, D16, 20673-20696.

6. P6, L6. If $\mathbf{W}$ is the covariance matrix then its diagonal elements are the squares of the standard errors in the measurements (not the errors themselves) and in equation (4) you should replace $\mathbf{W}^2$ with $\mathbf{W}$. If, instead, the diagonal elements of $\mathbf{W}$ are the errors then it is correct to keep $\mathbf{W}^2$ in eq. (4) but you should say that $\mathbf{W}^2$ (not $\mathbf{W}$) is the covariance matrix.

Ans.) We have adopted that $\mathbf{W}^2$ is the covariance matrix in the revised manuscript. (P6, L10-12).

7. P6, L15 and eq. (5) and (6). I do not find the notation very clear. Wouldn't it be better to just say that $\mathbf{y}^{mea}$ contains $\tau_{ext}^{mea}$ at all wavelengths, $II^{mea}$ at all wavelengths and scattering angles and $\delta_{ave}^{mea}$ at 532 nm? Or, if you want to use an equation, you may use the more established notation
$$\mathbf{y}^{mea} = [\tau_{ext}^{mea}(\lambda), \cdots\cdots, \tau_{ext}^{mea}(\lambda), I^{mea}(\Theta, \lambda), \cdots\cdots, I^{mea}(\Theta, \lambda), \delta_{ave}^{mea}(532 \text{ nm})]$$
or you can summarize the content of the vector in a table. Similarly, in eq. (6) wouldn't it be clearer if you just state that x contains real and imaginary part of refractive index at all SKYR wavelengths plus all the other parameters?

Ans.) We deleted the equations and just described the contents of the vectors $\mathbf{x}$ and $\mathbf{y}$ in the revised manuscript (P6, L19-22).

8. P9, equations. Again, the notation of the formulas looks a bit confusing to me. The suggestions I would give you are similar to those given in the previous point.

Ans.) Same as the above answer (P9, L10-15).

9. P11, eq.(21). $\mathbf{W}$ and $\mathbf{W_a}$ should be uppercase. Please re-check whether there should be $\mathbf{W}^2$ or $\mathbf{W}$ depending on the choice you make about the meaning of $\mathbf{W}$ (see point 6).

Ans.) We have checked it. We did not need any modifications.

10. P12, L24. "In actuality" -> "Actually" or "Eventually".

Ans.) We have corrected it (P13, L8).

11. P13, L27. It is not clear to me what does "the broadband wavelength regions of solar radiation" mean.

Ans.) We have revised it as "the solar wavelength region" (P13, L11).

12. P13, L28. "The wavelengths of the optical properties" -> "The wavelengths at which the optical properties . . . are obtained".

Ans.) We have corrected it (P13, L11-12).

13. P14, L2. Which kind of interpolation did you use to compute the refractive index between 532 and 1064 nm? I guess linear is enough for the real part, but what about the imaginary part?

Ans.) We used the linear interpolation in the log-log plane. We have added this explanation (P13, L13).

14. P15, L2. "other than" -> "except" (it sounds clearer).

Ans.) We have corrected it (P14, L15).

15. P15, L3. Is the size distribution only retrieved at 540 and 3540 m? Is that enough to call that a "profile of the aerosol size distribution"? Furthermore, you assume a bimodal lognormal size distribution and – if I understood correctly – you fit mode radii and widths for the entire column in step 1, whereas in step 2 you keep those values fixed but update $C_1$ and $C_2$ per layer. Then, I guess, your estimate of the aerosol size distribution profile is controlled by these two parameters, right? Wouldn't it be more informative to compare and plot the profiles of $C_1(z)$ and $C_2(z)$?

Ans.) It is difficult to calculate the vertical profiles of "true" $C_1$ and $C_2$ because more than three aerosol components are mixed externally in the sensitivity tests (see Table 1).
We showed the size distributions at only two altitudes, but we selected the representative altitudes for the transported and background aerosols. We think that is enough to call a "profile of the aerosol size distribution".

16. P15, L7. "plotted" -> "shows"; "the aerosol optical thickness of" -> "an aerosol optical thickness of".

Ans.) We have corrected them (P14, L1, 14, and 20, ).

17. P15, L8-9. You say the profiles of the refractive index were not estimated well, but doesn't that only apply to the imaginary part? You may want to specify that.

Ans.) We have revised the sentence (P14, L21-22).

18. P19, L21. SKR -> SKYR ?

Ans.) We have corrected the misprint (P19, L4).

[revised manuscript text omitted]

Our algorithm estimates the aerosol parameters, based on the maximum a posteriori (MAP) scheme. The aerosol parameters for the best fit to all of the measurements and a priori information are obtained by maximizing the posterior probability density function (PDF):

$$P(\mathbf{x}|\mathbf{y}^{mea}) = P(\mathbf{y}^{mea}|\mathbf{x})P(\mathbf{x})/P(\mathbf{y}^{mea}), \tag{3}$$

where vector $\mathbf{y}^{mea}$ describes the measurements, vector $\mathbf{x}$ describes the aerosol parameters to be estimated, $P(\mathbf{y}^{mea}|\mathbf{x})$ is the conditional PDF of $\mathbf{y}^{mea}$ given $\mathbf{x}$, $P(\mathbf{x})$ is the prior PDF of $\mathbf{x}$, and $P(\mathbf{y}^{mea})$ is the prior PDF of $\mathbf{y}^{mea}$. Since $P(\mathbf{y}^{mea})$ does not depend on $\mathbf{x}$, the PDF to be maximized is $P(\mathbf{y}^{mea}|\mathbf{x})P(\mathbf{x})$. Assuming the normal distribution, $P(\mathbf{y}^{mea}|\mathbf{x})P(\mathbf{x})$ is defined

5    as

$$P(\mathbf{y}^{mea}|\mathbf{x})P(\mathbf{x}) \propto \exp[-\frac{1}{2}\left(\mathbf{y}^{mea} - \mathbf{y}(\mathbf{x})\right)^T (\mathbf{W}^2)^{-1}\left(\mathbf{y}^{mea} - \mathbf{y}(\mathbf{x})\right)$$

$$-\frac{1}{2}(\mathbf{x} - \mathbf{x}_a)^T (\mathbf{W}_a^2)^{-1}(\mathbf{x} - \mathbf{x}_a)], \tag{4}$$

10   where vector $\mathbf{y}(\mathbf{x})$ comprise the values corresponding to $\mathbf{y}^{mea}$ calculated from $\mathbf{x}$ by the forward model, and matrix $\mathbf{W}^2$ is the covariance matrix of $\mathbf{y}$ and is assumed to be diagonal in this study. The diagonal elements of $\mathbf{W}$ are the standard errors in the measurements. The vector $\mathbf{x}_a$ comprises the a priori value of $\mathbf{x}$, and matrix $\mathbf{W}_a$ is the associated covariance matrix. The maximum of Eq. (4) is obtained by minimizing the objective function,

$$f(\mathbf{x}) = \frac{1}{2}(\mathbf{y}^{mea} - \mathbf{y}(\mathbf{x}))^T (\mathbf{W}^2)^{-1}(\mathbf{y}^{mea} - \mathbf{y}(\mathbf{x})) + \frac{1}{2}(\mathbf{x} - \mathbf{x}_a)^T (\mathbf{W}_a^2)^{-1}(\mathbf{x} - \mathbf{x}_a). \tag{5}$$

We search for the best $\mathbf{x}$, which minimizes $f(\mathbf{x})$, by iterations of the Gauss-Newton method with a line search, $\mathbf{x}_{i+1} = \mathbf{x}_i + \alpha_j \mathbf{d}_i$ ("Update $\mathbf{x}$" in Fig. 1). This minimization procedure is described in Sects 2.1.4 and 2.1.5.

In step 1, the vector $\mathbf{y}^{mea}$ consists of $\tau_{ext}^{mea}(\lambda)$ at SKYR wavelengths, $I^{mea}(\Theta, \lambda)$ at SKYR wavelengths and scattering

20   angles, and $\delta_{ave}^{mea}$ at 532 nm. Vector $\mathbf{x}$ describes the aerosol parameters and consists of the real and imaginary parts of the refractive index at SKYR wavelengths ($n(\lambda)$ and $k(\lambda)$), and the parameters for the size distribution ($C_1, C_2, r_{m,1}, r_{m,2}, s_1, s_2,$ and $\varepsilon$). We assumed the following bi-modal lognormal size distribution,

$$\frac{dV(r)}{d\ln r} = \sum_{i=1}^{2} \frac{dV_i(r)}{d\ln r} = \sum_{i=1}^{2} \frac{C_i}{\sqrt{2\pi}s_i} \exp\left[-\frac{1}{2}\left(\frac{\ln r - \ln r_{m,i}}{s_i}\right)^2\right], \tag{6}$$

where $C_i$, $r_{m,i}$, and $s_i$ are volume, radius, and width, respectively of the fine ($i = 1$) and coarse ($i = 2$) modes. $\varepsilon$ is the volume ratio of non-spherical particles to total particles in the coarse mode.

We constructed the forward models $\mathbf{y}(\mathbf{x})$ to calculate $\tau_{ext}(\lambda)$, $I(\Theta, \lambda)$, and $\delta_{ave}(532nm)$ from the above-mentioned aerosol parameters. The optical properties of aerosols were calculated by a method similar to that of Lopatin et al. (2013) as follows:

$$\tau_{\frac{ext}{sca}}(\lambda) = \sum_k \frac{dV_1(r_k)}{d\ln r} K^S_{\frac{ext}{sca}}(\lambda, n, k, r_k) + \sum_k (1-\varepsilon) \frac{dV_2(r_k)}{d\ln r} K^S_{\frac{ext}{sca}}(\lambda, n, k, r_k)$$

$$+ \sum_k \varepsilon \frac{dV_2(r_k)}{d\ln r} K^{NS}_{ext/sca}(\lambda, n, k, r_k), \tag{7}$$

$$\tau_{sca}(\lambda) P_{ii}(\Theta, \lambda) = \sum_k \frac{dV_1(r_k)}{d\ln r} K^S_{ii}(\Theta, \lambda, n, k, r_k)$$

$$+ \sum_k (1-\varepsilon) \frac{dV_2(r_k)}{d\ln r} K^S_{ii}(\Theta, \lambda, n, k, r_k) + \sum_k \varepsilon \frac{dV_2(r_k)}{d\ln r} K^{NS}_{ii}(\Theta, \lambda, n, k, r_k), \tag{8}$$

where $\tau_{ext/sca}(\lambda)$ denotes the optical thickness for extinction and scattering, and $\tau_{sca}(\lambda) P_{ii}(\Theta, \lambda)$ denotes the directional scattering corresponding to the scattering matrix elements $P_{ii}(\Theta, \lambda)$. $K^S_{...}$ and $K^{NS}_{...}$ are the kernels of extinction and scattering properties for spherical and non-spherical particles, respectively. The kernel for spherical particles was constructed by using Mie theory. The kernel for non-spherical particles was constructed by using the data table of Dubovik et al. (2006), which assumes randomly oriented polydisperse spheroids with a fixed aspect ratio distribution for mineral dust.

$I(\Theta, \lambda)$ is computed by the radiative transfer code in the SKYRAD.PACK ver. 4.2 (Nakajima et al., 1996). Although only one atmospheric layer is considered by SKYRAD.PACK, we assumed that the atmosphere consists of two layers. The bottom layer includes aerosols, and its top altitude $z_{max}$ is determined by lidar measurement. The upper layer is aerosol-free. The Rayleigh scattering is calculated by the method of Bucholtz (1995). The vertical ozone profile is approximated by the formula of Green (1964), and the ozone absorption coefficient is adopted from the LOWTRAN 7 database (Kneizys et al., 1988).

$\delta_{ave}(\lambda)$ is calculated as

$$\delta_{ave}(\lambda) = \left( \frac{\beta_m(\lambda)\delta_m(\lambda)}{1+\delta_m(\lambda)} + \frac{\beta_p(\lambda)\delta_p(\lambda)}{1+\delta_p(\lambda)} \right) \Big/ \left( \frac{\beta_m(\lambda)}{1+\delta_m(\lambda)} + \frac{\beta_p(\lambda)}{1+\delta_p(\lambda)} \right), \tag{9}$$

$$\beta_{p/m}(\lambda) = \tau_{sca,p/m}(\lambda)P_{11,p/m}(180^\circ, \lambda), \tag{10}$$

$$\delta_{p/m}(\lambda) = \frac{1 - P_{22,p/m}(180^\circ, \lambda)/P_{11,p/m}(180^\circ, \lambda)}{1 + P_{22,p/m}(180^\circ, \lambda)/P_{11,p/m}(180^\circ, \lambda)}, \tag{11}$$

where $\beta_{p/m}(\lambda)$, $\delta_{p/m}(\lambda)$, and $\tau_{sca,p/m}(\lambda)$ are the backscatter coefficient, the depolarization ratio, and the scattering optical thickness, respectively, for particulate (*p*) and molecular (*m*) scattering in the bottom aerosol layer.

The first term of Eq. (5) is calculated by using the above-mentioned $\mathbf{y}^{mea}$ and $\mathbf{y}(\mathbf{x})$. The values of the diagonal matrix $\mathbf{W}$ are the measurement errors, which were assumed to be 0.01 for $\tau_{ext}^{mea}(\lambda \geq 500\text{nm})$ and 0.02 for $\tau_{ext}^{mea}(\lambda < 500\text{nm})$, 5 % for
10   $I^{mea}(\theta, \lambda)$, and 20 % for $\delta_{ave}^{mea}(532 \text{ nm})$.

To reduce the effects of measurement errors on retrievals, Dubovik and King (2000) restricted the spectral variability of the refractive index by limiting the length of the derivatives of the refractive index with respect to the wavelength. They considered this a priori smoothness constraint as being of the same nature as a measurement and incorporated the smoothness constraint into their retrieval scheme. We adopted this consideration and introduced the smoothness constraint
15   for the real and imaginary parts of the refractive index. The first derivatives of the refractive index with respect to the wavelengths are defined as

$$\mathbf{y}_a(\boldsymbol{x}) = \left( \frac{\ln n(\lambda_1) - \ln n(\lambda_2)}{\ln \lambda_1 - \ln \lambda_2} \quad \cdots \quad \frac{\ln n(\lambda_6) - \ln n(\lambda_7)}{\ln \lambda_6 - \ln \lambda_7} \quad \frac{\ln k(\lambda_1) - \ln k(\lambda_2)}{\ln \lambda_1 - \ln \lambda_2} \quad \cdots \quad \frac{\ln k(\lambda_6) - \ln k(\lambda_7)}{\ln \lambda_6 - \ln \lambda_7} \right). \tag{12}$$

20   $\mathbf{y}_a(\boldsymbol{x})$ is incorporated into $\mathbf{y}(\mathbf{x})$, and the value of $\mathbf{y}^{mea}$ corresponding to $\mathbf{y}_a(\boldsymbol{x})$ is zero. The values entered in the weight matrix $\mathbf{W}$ were 0.2 for the real part and 1.25 for the imaginary part. These values are used in the AERONET retrieval (Dubovik and King, 2000) for constraining the spectral variability of the refractive index to some practically reasonable ranges.

[revised manuscript text omitted]

In step 2, the number of estimated parameters is larger than the number of measurements, so the lidar measurements would be insufficient for retrieving unique solutions of the refractive index even if the columnar properties obtained in step 1 are added to $\mathbf{y}^{mea}$. Therefore, we gave a priori constraints to the real and imaginary parts of the refractive index using the step 1 results $k^{s1}(\lambda)$ and $\tau_{ext}^{s1}(\lambda)$. Assuming the log-normal PDF for $P(\mathbf{x})$, the second term of Eq. (5) is replaced by

$$\frac{1}{2}(\ln\mathbf{n} - \ln\mathbf{n}_a^{s1})^T(\mathbf{W}_a^2)^{-1}(\ln\mathbf{n} - \ln\mathbf{n}_a^{s1}) + \frac{1}{2}(\ln\mathbf{k} - \ln\mathbf{k}_a^{s1})^T(\mathbf{W}_a^2)^{-1}(\ln\mathbf{k} - \ln\mathbf{k}_a^{s1}).\qquad (15)$$

The matrix $\mathbf{W}_a$, which determines the strength of the constraints, is assumed to be diagonal and the values are obtained by a method similar to that of Dubovik and King (2000). The possible variability ranges of the refractive index for aerosols are from 1.33 to 1.6 for the real part, and from 0.0005 to 0.5 for the imaginary part. We considered these intervals to be 68 % confidence intervals, $[\ln n^{s1} - \Delta n, \ln n^{s1} + \Delta n]$ and $[\ln k^{s1} - \Delta k, \ln k^{s1} + \Delta k]$, and determined the weight values as

$$W_a = \begin{cases} 0.5(\ln n_{max} - \ln n_{min}), & \text{for real part} \\ 0.5(\ln k_{max} - \ln k_{min}), & \text{for imaginary part} \end{cases}.\qquad (16)$$

The objective function is minimized by the procedures described in Sects 2.1.4 and 2.1.5, and the vertical profiles of the real and imaginary parts of the refractive index ($n(\lambda, z)$, $k(\lambda, z)$) at MIEL wavelengths, the size distribution ($C_1(z)$, $C_2(z)$), and the volume ratio of non-spherical particles to total particles in the coarse mode ($\varepsilon(z)$) are optimized. Finally, the vertical profiles of the refractive index, size distribution ($\frac{dV(r,z)}{d\ln r}$), extinction coefficients ($\sigma_{ext}(\lambda, z)$), single-scattering albedo ($\omega_0(\lambda, z)$), and asymmetry parameter ($g(\lambda, z)$) are output. The wavelengths of these optical properties are 532 and 1064 nm for Type 1 and 2 data sets, and 355, 532, 1064 nm for Type 3 data set.

**2.1.4 Minimization procedure**

In both steps 1 and 2, $\mathbf{x}$ was optimized to $\mathbf{y}^{mea}$ by finding the minimum of $f(\mathbf{x})$ in $\mathbf{x}$ space. We employed the Gauss-Newton method to search for the minimum. The Gauss-Newton method searches for the minimum by iteratively updating $\mathbf{x}$, and it is usually combined with a line search method (Nocedal and Wright, 2006). In this procedure (Fig. 1c), $\mathbf{x}$ is updated by $\mathbf{x}_{i+1} = \mathbf{x}_i + \alpha_j \mathbf{d}_i$, where the value of $f(\mathbf{x})$ in the $\mathbf{x}$ space decreases in the vector $\mathbf{d}_i$ direction, and $\alpha_j$ is a positive parameter that minimizes $f(\mathbf{x})$ in direction $\mathbf{d}_i$. $\mathbf{d}_i$ is determined by the Gauss-Newton method in the outer-loop, and $\alpha_j$ is determined by a line search in the inner loop. $\mathbf{d}_i$ is obtained by solving the normal equation,

$$[\mathbf{J}(\mathbf{x}_i)^T (\mathbf{w}^2)^{-1} \mathbf{J}(\mathbf{x}_i) + (\mathbf{w}_a^2)^{-1}]\mathbf{d}_i = \mathbf{J}(\mathbf{x}_i)^T (\mathbf{w}^2)^{-1}(\mathbf{y}^{mea} - \mathbf{y}(\mathbf{x}_i))$$
$$+ (\mathbf{w}_a^2)^{-1}(\mathbf{x}_i - \mathbf{x}_a), \tag{17}$$

[revised manuscript text omitted]

---

## Referee Comment (RC3) · Anonymous Referee #2 · 6 Jun 2016

This article describes an algorithm to retrieve aerosol properties from a combination of ground-based lidar and sun-photometer data. Authors suggest a novel approach, which suggests retrieving vertical profile of aerosol complex refractive index, which distinguishes the mentioned SKYLIDAR algorithm from other similar approaches. Also paper addresses the issue of computation of solar heating rate profiles based on the results of such combined retrievals. Mentioned issues are in a high interest of scientific community, and correspond well the scope of the AMT. Proposed approach was throughoutly tested on the several sets of synthetic data, in noisy and noise free conditions, applied to the real measurements with a proper comparison of the retrieved results with other methods.

[Figure]

The paper is nice to read; compact and informative description of the work. However, I think that authors should've put more efforts to properly illustrate some of their results.

I think paper could be published after minor changes. Below I provide some issues that, in my opinion, should be resolved to improve the publication.

Technical comments:

Page 3 line 6. "profiles" instead of "profile"

Page 7 line 6. Although Lopatin et al. uses in his work spheroid model it wasn't author's personal contribution, I think a paper of Dubovik et al, 2006 on spheroids more corresponds to this reference.

Page 11 line 13. "are output" , doesn't sound like a proper sentence to me, consider adding additional verb, for e.g. "are produced at(as) output"

Page 12 line 23. "Eq.(23) can", maybe it'll be better to replace it with "can(could) be" to maintain the passive voice like in the first part of the sentence.

Page 14 line 19. "The estimated real" I think adjective should be changed to "assumed" or "modelled", since it is hard to tell what was expected and what was retrieved.

Page 16 line 17 & Page 17 line 8. Please, indicate the wavelength of the single scattering albedo at which the comparison was made. And were the wavelengths used for comparison the same?

Page 18 line 29. I think that vertical profile of spherical/non-spherical particles ratio was forgotten.

Page 19 line 21. "SKYR" instead of "SKR"

General comments:

Page 9 line 2. I would like to see more info on extrapolation, first the method and secondly it is not clear if optical properties (optical thickness and single scattering albedo)

were interpolated too, or they were calculated following eq. 8,9 on the base of inter-extrapolated values of n and k.

Page 13 lines 6-14. It is not clear, how the transforming function was used. Was look-up-table used to overcome problems with possible zero and infinity values? If yes, what are the values of the LUT at 1.3 and 1.6? Also it is not clear if max and min values are actually reachable during retrieval (exact values). Please, provide more information on this in the text.

Page 14. I would like to see how the "TRUE" vertical profiles of refractive index were estimated. It's not straightforward how to get them from the mixtures of components with different n, k and vertical distribution profiles provided in the Table 1.

Page 14 line 25. Please, indicate which type of random noise distribution was used.

Page 15 lines 13-24. I would like to see a bit more of analysis of the sensitivity test results. Not only state the facts, but also add some discussion why in your opinion this happened:

1. "vertical profiles of refractive index and single scattering albedo of dust were re-produced well, but not those of transported pollution ". Could you provide a possible explanation why?

2. "The retrieval with and without HSRL data did not differ". Could you provide at least some explanation why there is no sensitivity to additional data for all types of aerosol?

3. Concerning oscillations, I think somehow it should be indicated that other methods like Chaikovsky et al, Lopatin et al. use additional smoothness constraints on aerosol vertical distributions to overcome observed oscillations. Maybe plan for implementation of such feature in the future work should be placed in conclusions.

4. At lines 13-14 it is claimed that size accurate retrieval of size distribution is enough to get extinction and asymmetry factor, but in the introduction (page 3 lines 6-7) it is indicated that refractive index reproduction is crucial to get asymmetry factor. It

is a bit misleading; consider reformulating one of the phrases not to have a logical contradiction. Maybe a more general (for all aerosol types) conclusion about vertical refractive index retrieval and how it influences retrievals of optical parameters should be done, since it is a feature that distinguishes SKYLIDAR from other lidar-sunphotometer combination approaches.

Page 17. Comparisons of columnar properties. I think a brief description of the differences of first step of SKYLIDAR and SKYRAD.PACK is needed, since not everyone is well informed about SKYRAD.

Page 18. Summary It would be nice to see a list of future developments/improvements or plans for studies in the conclusions section. This will emphasize the methodology in the work: made tests>made conclusions>plans for further improvement.

Table 1. Could you add spherical/non-spherical to the description? And are dust and insoluble particles were considered to be 100% non-spherical? If not, please, provide a value.

Figures 4-9. My strongest belief that figures do not provide convincing illustrations that parameters are "retrieved well" as it was frequently stated in most of descriptions of the figures. I have several suggestions how this could be significantly improved:

1. It is really hard to analyze retrieval accuracy on the plot that is a bit bigger than a mail stamp. I suggest making the graphs bigger. I assume authors made plots that small to fit everything, so my suggestion will be to split every figure in two, showing physical and optical properties (extinction, ssa, and asymmetry factor) separately. 2. Statement that "vertical profiles of size distribution were estimated well" is supported by illustration of the SDs only at two altitudes! I suggest to put assumed and retrieved vertical profiles of C1 and C2. In my opinion it'll be much more illustrative and convincing. 3. I'm eager to see also the vertical profile of epsilon (spherical/non-spherical aerosol fraction), both modeled and retrieved.

To sum the desired changes in graphs, I would like to see bigger plots that show all the parameters of aerosol model that are retrieved by the SKYLIDAR directly compared to the data that was used to produce the synthetic dataset. In my opinion these vertical profiles will be much more convincing and easy to understand, than trying to analyze the contents of Table 1. I hope same method that was used by authors to provide resulting vertical profiles for n and k, could be also applied to provide profiles of C1, C2 and epsilon.

---

## Author Comment (AC2) · 17 Jun 2016

Reply to Referee 2 comment

The authors thank the referee for a though reading of the manuscript and helpful comments. We have responded to the all comments, namely.

The changes to meet your comments are marked with green background in the revised manuscript.

Technical comments:

Page 3 line 6. "profiles" instead of "profile"
Ans.) We have corrected it (Page 3 line 6).

Page 7 line 6. Although Lopatin et al. uses in his work spheroid model it wasn't author's personal contribution, I think a paper of Dubovik et al, 2006 on spheroids more corresponds to this reference.
Ans.) We have changed it to "Dubovik et al. (2006)" (Page 7 line 4).

Page 11 line 13. "are output" , doesn't sound like a proper sentence to me, consider adding additional verb, for e.g. "are produced at(as) output"
Ans.) We have corrected it (Page 10 line 23).

Page 12 line 23. "Eq.(23) can", maybe it'll be better to replace it with "can(could) be" to maintain the passive voice like in the first part of the sentence.
Ans.) We have corrected it (Page 12 line 8).

Page 14 line 19. "The estimated real" I think adjective should be changed to "assumed" or "modelled", since it is hard to tell what was expected and what was retrieved.
Ans.) The sentence described the retrieval results of the refractive index. We have revised the sentence as a simple explanation, "The retrieval errors of the refractive index were small, except for the real part at 355 nm.". (Page 14 lines 9-10)

Page 16 line 17 & Page 17 line 8. Please, indicate the wavelength of the single scattering albedo at which the comparison was made. And were the wavelengths

used for comparison the same?

Ans.) We have added the wavelength of the single scattering albedo in the revised manuscript (Page 17 line 9, and 29). The wavelength of our retrieved single scattering albedo is 532 nm, but the wavelengths used in the comparisons are 550nm (Page 17 line 9), and 440 and 675 nm (Page 17 line 29). Our discussions are not influenced by these differences.

Page 18 line 29. I think that vertical profile of spherical/non-spherical particles ratio was forgotten.

Ans.) Yes. We have added it (Page 19 line 23).

Page 19 line 21. "SKYR" instead of "SKR"

Ans.) We have corrected it (Page 20 line 21).

General comments:

Page 9 line 2. I would like to see more info on extrapolation, first the method and secondly it is not clear if optical properties (optical thickness and single scattering albedo) were interpolated too, or they were calculated following eq. 8, 9 on the base of inter-extrapolated values of n and k.

Ans.) The linear interpolation and extrapolation in the log–log plot were used for estimating n and k at lidar wavelengths. The optical thickness and single scattering albedo at lidar wavelengths are then calculated from those using eq. 8 and 9. We have added this explanation (Page 9 lines 1-3).

Page 13 lines 6-14. It is not clear, how the transforming function was used. Was lookup-table used to overcome problems with possible zero and infinity values? If yes, what are the values of the LUT at 1.3 and 1.6? Also it is not clear if max and min values are actually reachable during retrieval (exact values). Please, provide more information on this in the text.

Ans.) The LUTs of $K_{ext/scat}$ and $K_{ii}$ in eq. 7 and 8 for different values of the size parameter and refractive index were used for the rapid calculation in this study. The minimum and maximum values of the refractive index are 1.33 and 1.6 for the real part, 0.0005 and 0.5 for the imaginary part, respectively.

There is a possibility that $x_{i+1}$ excesses the minimum or maximum values of the

refractive index in the iteration of $x_{i+1} = x_i + \alpha_j d_i$. For example, this can happen due to the large noise of the lidar data. Our transformation prevents this problem. The volume ratio of the non-spherical particles to total particles also has the minimum (0) and maximum values (1) by the physical reason. The ratio easily reaches to the minimum and maximum values in the optically thin case (Fig. 6) The transformations for x and y are applied to Eq. 4, and then the minimization of the cost function starts.

We have added more information in the revised manuscript (Page 12 line 17 to Page 13 line 3).

Page 14. I would like to see how the "TRUE" vertical profiles of refractive index were estimated. It's not straightforward how to get them from the mixtures of components with different n, k and vertical distribution profiles provided in the Table 1.

Ans.) The "TRUE" values are calculated by the weighted arithmetic mean with the scattering coefficient of each component (Page 14 lines 7-8).

Page 14 line 25. Please, indicate which type of random noise distribution was used.

Ans.) We used the uniform distribution (Page 14 lines 19-21).

Page 15 lines 13-24. I would like to see a bit more of analysis of the sensitivity test results. Not only state the facts, but also add some discussion why in your opinion this happened:

Ans.) We have added the more discussions (Page 14 line 6 to Page 16 line 14).

1. "vertical profiles of refractive index and single scattering albedo of dust were reproduced well, but not those of transported pollution ". Could you provide a possible explanation why?

Ans.) We investigated the dependencies of the extinction-to backscatter ratio on the refractive index, and the mode radius of the mono-modal size distribution (see Fig. 16, and Page 15 line 24 to Page 16 line 4.). The extinction-to backscatter ratio was sensitive to the real and imaginary parts in the case that the coarse mode was dominant, but not in the case that the fine mode was dominant. Therefore, the vertical profiles of the refractive index for the transported pollution were not reproduced.

2. "The retrieval with and without HSRL data did not differ". Could you provide at least some explanation why there is no sensitivity to additional data for all types of aerosol?

Ans.) The advantage of HSRL data is that particle backscatter and extinction coefficients are obtained separately. Our algorithm in the step 1 also estimates the optical thickness and the other parameters related to the backscatter coefficient separately. The retrievals of the step 1 are the columnar values, but manipulating the MIEL data with the step 1 retrievals would have an effect similar to the addition of HSRL data. We have added this discussion (Page 16 lines 4-8).

3. Concerning oscillations, I think somehow it should be indicated that other methods like Chaikovsky et al, Lopatin et al. use additional smoothness constraints on aerosol vertical distributions to overcome observed oscillations. Maybe plan for implementation of such feature in the future work should be placed in conclusions.

Ans.) The large oscillations of the single-scattering albedo and asymmetry factor in the optically thin case were caused by the large oscillations of the volume concentrations (Fig. 6). The smoothness constraint for the vertical variation of the volume concentration has a possibility to improve the retrievals in the optically thin case. We will introduce the constraint to the algorithm in the future (Page 16 lines 9-14).

4. At lines 13-14 it is claimed that size accurate retrieval of size distribution is enough to get extinction and asymmetry factor, but in the introduction (page 3 lines 6-7) it is indicated that refractive index reproduction is crucial to get asymmetry factor. It is a bit misleading; consider reformulating one of the phrases not to have a logical contradiction. Maybe a more general (for all aerosol types) conclusion about vertical refractive index retrieval and how it influences retrievals of optical parameters should be done, since it is a feature that distinguishes SKYLIDAR from other lidar-sunphotometer combination approaches.

Ans.) We added the dependencies of optical properties on physical properties and explained the general features in the revised manuscript (Page 15 lines 14-22).

Page 17. Comparisons of columnar properties. I think a brief description of the differences of first step of SKYLIDAR and SKYRAD.PACK is needed, since not everyone is well informed about SKYRAD.

Ans.) We have added the description of the difference (Page 18 lines 18-23).

Page 18. Summary It would be nice to see a list of future developments/improvements or plans for studies in the conclusions section. This will emphasize the methodology in the work: made tests>made conclusions>plans for further improvement.
Ans.) Yes. We have a plan to analyze the long-term measurements of SKYR and MIEL in the East Asian region. And, we are now developing the method to estimate the aerosol vertical profile from the space-borne lidar and imager data using our developed optimization techniques. We have added these in the summary (Page 20 lines 17-26).

Table 1. Could you add spherical/non-spherical to the description? And are dust and insoluble particles were considered to be 100% non-spherical? If not, please, provide a value.
Ans.) We assumed that the dust and insoluble particles are 100 % non-spherical. We have added spherical/non-spherical to Table 1.

Figures 4-9. My strongest belief that figures do not provide convincing illustrations that parameters are "retrieved well" as it was frequently stated in most of descriptions of the figures. I have several suggestions how this could be significantly improved:
1. It is really hard to analyze retrieval accuracy on the plot that is a bit bigger than a mail stamp. I suggest making the graphs bigger. I assume authors made plots that small to fit everything, so my suggestion will be to split every figure in two, showing physical and optical properties (extinction, ssa, and asymmetry factor) separately. 2. Statement that "vertical profiles of size distribution were estimated well" is supported by illustration of the SDs only at two altitudes! I suggest to put assumed and retrieved vertical profiles of C1 and C2. In my opinion it'll be much more illustrative and convincing. 3. I'm eager to see also the vertical profile of epsilon (spherical/non-spherical aerosol fraction), both modeled and retrieved.
To sum the desired changes in graphs, I would like to see bigger plots that show all the parameters of aerosol model that are retrieved by the SKYLIDAR directly compared to the data that was used to produce the synthetic dataset. In my opinion these vertical profiles will be much more convincing and easy to understand, than trying to analyze the contents of Table 1. I hope same method that was used by

authors to provide resulting vertical profiles for n and k, could be also applied to provide profiles of C1, C2 and epsilon.

Ans.) We agreed your comments and separately created the figures of physical and optical properties. We added figures of C1, C2, and epsilon to the revised manuscript. The explanations and discussions for new figures are added to the revised manuscript (Page 14 line 6 to Page 16 line 14).

[revised manuscript text omitted]

Our algorithm estimates the aerosol parameters, based on the maximum a posteriori (MAP) scheme. The aerosol parameters for the best fit to all of the measurements and a priori information are obtained by maximizing the posterior probability density function (PDF):

$$P(\mathbf{x}|\mathbf{y}^{mea}) = P(\mathbf{y}^{mea}|\mathbf{x})P(\mathbf{x})/P(\mathbf{y}^{mea}), \tag{3}$$

where vector $\mathbf{y}^{mea}$ describes the measurements, vector $\mathbf{x}$ describes the aerosol parameters to be estimated, $P(\mathbf{y}^{mea}|\mathbf{x})$ is the conditional PDF of $\mathbf{y}^{mea}$ given $\mathbf{x}$, $P(\mathbf{x})$ is the prior PDF of $\mathbf{x}$, and $P(\mathbf{y}^{mea})$ is the prior PDF of $\mathbf{y}^{mea}$. Since $P(\mathbf{y}^{mea})$ does not depend on $\mathbf{x}$, the PDF to be maximized is $P(\mathbf{y}^{mea}|\mathbf{x})P(\mathbf{x})$. Assuming the normal distribution, $P(\mathbf{y}^{mea}|\mathbf{x})P(\mathbf{x})$ is defined as

$$P(\mathbf{y}^{mea}|\mathbf{x})P(\mathbf{x}) \propto \exp[-\frac{1}{2}\left(\mathbf{y}^{mea} - \mathbf{y}(\mathbf{x})\right)^T (\mathbf{W}^2)^{-1}\left(\mathbf{y}^{mea} - \mathbf{y}(\mathbf{x})\right)$$

$$-\frac{1}{2}(\mathbf{x} - \mathbf{x}_a)^T (\mathbf{W}_a^2)^{-1}(\mathbf{x} - \mathbf{x}_a)], \tag{4}$$

where vector $\mathbf{y}(\mathbf{x})$ comprise the values corresponding to $\mathbf{y}^{mea}$ calculated from $\mathbf{x}$ by the forward model, and matrix $\mathbf{W}^2$ is the covariance matrix of $\mathbf{y}$ and is assumed to be diagonal in this study. The diagonal elements of $\mathbf{W}$ are the standard errors in the measurements. The vector $\mathbf{x}_a$ comprises the a priori value of $\mathbf{x}$, and matrix $\mathbf{W}_a$ is the associated covariance matrix. The maximum of Eq. (4) is obtained by minimizing the objective function,

$$f(\mathbf{x}) = \frac{1}{2}\left(\mathbf{y}^{mea} - \mathbf{y}(\mathbf{x})\right)^T (\mathbf{W}^2)^{-1}\left(\mathbf{y}^{mea} - \mathbf{y}(\mathbf{x})\right) + \frac{1}{2}(\mathbf{x} - \mathbf{x}_a)^T (\mathbf{W}_a^2)^{-1}(\mathbf{x} - \mathbf{x}_a). \tag{5}$$

We search for the best $\mathbf{x}$, which minimizes $f(\mathbf{x})$, by iterations of the Gauss-Newton method with a line search, $\mathbf{x}_{i+1} = \mathbf{x}_i + \alpha_j \mathbf{d}_i$ ("Update $\mathbf{x}$" in Fig. 1). This minimization procedure is described in Sects 2.1.4 and 2.1.5.

In step 1, the vector $\mathbf{y}^{mea}$ consists of $\tau_{ext}^{mea}(\lambda)$ at SKYR wavelengths, $I^{mea}(\Theta, \lambda)$ at SKYR wavelengths and scattering angles, and $\delta_{ave}^{mea}$ at 532 nm. Vector $\mathbf{x}$ describes the aerosol parameters and consists of the real and imaginary parts of the refractive index at SKYR wavelengths ($n(\lambda)$ and $k(\lambda)$), and the parameters for the size distribution ($C_1$, $C_2$, $r_{m,1}$, $r_{m,2}$, $s_1$, $s_2$, and $\varepsilon$). We assumed the following bi-modal lognormal size distribution,

$$\frac{dV(r)}{d\ln r} = \sum_{i=1}^{2} \frac{dV_i(r)}{d\ln r} = \sum_{i=1}^{2} \frac{C_i}{\sqrt{2\pi}s_i} \exp\left[-\frac{1}{2}\left(\frac{\ln r - \ln r_{m,i}}{s_i}\right)^2\right], \tag{6}$$

where $C_i$, $r_{m,i}$, and $s_i$ are volume, radius, and width, respectively of the fine ($i = 1$) and coarse ($i = 2$) modes. $\varepsilon$ is the volume ratio of non-spherical particles to total particles in the coarse mode.

We constructed the forward models $\mathbf{y}(\mathbf{x})$ to calculate $\tau_{ext}(\lambda)$, $I(\Theta, \lambda)$, and $\delta_{ave}(532nm)$ from the above-mentioned aerosol parameters. The optical properties of aerosols were calculated by a method similar to that of Dubovik et al. (2006) as follows:

$$\tau_{\underset{sca}{ext}}(\lambda) = \sum_k \frac{dV_1(r_k)}{d\ln r} K^S_{\underset{sca}{ext}}(\lambda, n, k, r_k) + \sum_k (1 - \varepsilon) \frac{dV_2(r_k)}{d\ln r} K^S_{\underset{sca}{ext}}(\lambda, n, k, r_k)$$

$$+ \sum_k \varepsilon \frac{dV_2(r_k)}{d\ln r} K^{NS}_{ext/sca}(\lambda, n, k, r_k), \tag{7}$$

$$\tau_{sca}(\lambda) P_{ii}(\Theta, \lambda) = \sum_k \frac{dV_1(r_k)}{d\ln r} K^S_{ii}(\Theta, \lambda, n, k, r_k)$$

$$+ \sum_k (1 - \varepsilon) \frac{dV_2(r_k)}{d\ln r} K^S_{ii}(\Theta, \lambda, n, k, r_k) + \sum_k \varepsilon \frac{dV_2(r_k)}{d\ln r} K^{NS}_{ii}(\Theta, \lambda, n, k, r_k), \tag{8}$$

where $\tau_{ext/sca}(\lambda)$ denotes the optical thickness for extinction and scattering, and $\tau_{sca}(\lambda) P_{ii}(\Theta, \lambda)$ denotes the directional scattering corresponding to the scattering matrix elements $P_{ii}(\Theta, \lambda)$. $K^S_{...}$ and $K^{NS}_{...}$ are the kernels of extinction and scattering properties for spherical and non-spherical particles, respectively. The kernel for spherical particles was constructed by using Mie theory. The kernel for non-spherical particles was constructed by using the data table of Dubovik et al. (2006), which assumes randomly oriented polydisperse spheroids with a fixed aspect ratio distribution for mineral dust.

$I(\Theta, \lambda)$ is computed by the radiative transfer code in the SKYRAD.PACK ver. 4.2 (Nakajima et al., 1996). Although only one atmospheric layer is considered by SKYRAD.PACK, we assumed that the atmosphere consists of two layers. The bottom layer includes aerosols, and its top altitude $z_{max}$ is determined by lidar measurement. The upper layer is aerosol-free. The Rayleigh scattering is calculated by the method of Bucholtz (1995). The vertical ozone profile is approximated by the formula of Green (1964), and the ozone absorption coefficient is adopted from the LOWTRAN 7 database (Kneizys et al., 1988).

$\delta_{ave}(\lambda)$ is calculated as

$$\delta_{ave}(\lambda) = \left( \frac{\beta_m(\lambda) \delta_m(\lambda)}{1 + \delta_m(\lambda)} + \frac{\beta_p(\lambda) \delta_p(\lambda)}{1 + \delta_p(\lambda)} \right) \Bigg/ \left( \frac{\beta_m(\lambda)}{1 + \delta_m(\lambda)} + \frac{\beta_p(\lambda)}{1 + \delta_p(\lambda)} \right), \tag{9}$$

$$\beta_{p/m}(\lambda) = \tau_{sca,p/m}(\lambda)P_{11,p/m}(180^\circ,\lambda), \tag{10}$$

$$\delta_{p/m}(\lambda) = \frac{1 - P_{22,p/m}(180^\circ,\lambda)/P_{11,p/m}(180^\circ,\lambda)}{1 + P_{22,p/m}(180^\circ,\lambda)/P_{11,p/m}(180^\circ,\lambda)}, \tag{11}$$

where $\beta_{p/m}(\lambda)$, $\delta_{p/m}(\lambda)$, and $\tau_{sca,p/m}(\lambda)$ are the backscatter coefficient, the depolarization ratio, and the scattering optical thickness, respectively, for particulate (*p*) and molecular (*m*) scattering in the bottom aerosol layer.

The first term of Eq. (5) is calculated by using the above-mentioned $\mathbf{y}^{mea}$ and $\mathbf{y}(\mathbf{x})$. The values of the diagonal matrix $\mathbf{W}$ are the measurement errors, which were assumed to be 0.01 for $\tau_{ext}^{mea}(\lambda \geq 500\text{nm})$ and 0.02 for $\tau_{ext}^{mea}(\lambda < 500\text{nm})$, 5 % for

10  $I^{mea}(\theta,\lambda)$, and 20 % for $\delta_{ave}^{mea}(532 \text{ nm})$.

To reduce the effects of measurement errors on retrievals, Dubovik and King (2000) restricted the spectral variability of the refractive index by limiting the length of the derivatives of the refractive index with respect to the wavelength. They considered this a priori smoothness constraint as being of the same nature as a measurement and incorporated the smoothness constraint into their retrieval scheme. We adopted this consideration and introduced the smoothness constraint

15  for the real and imaginary parts of the refractive index. The first derivatives of the refractive index with respect to the wavelengths are defined as

$$\mathbf{y}_a(\mathbf{x}) = \left( \frac{\ln n(\lambda_1) - \ln n(\lambda_2)}{\ln\lambda_1 - \ln\lambda_2} \quad \cdots \quad \frac{\ln n(\lambda_6) - \ln n(\lambda_7)}{\ln\lambda_6 - \ln\lambda_7} \quad \frac{\ln k(\lambda_1) - \ln k(\lambda_2)}{\ln\lambda_1 - \ln\lambda_2} \quad \cdots \quad \frac{\ln k(\lambda_6) - \ln k(\lambda_7)}{\ln\lambda_6 - \ln\lambda_7} \right). \tag{12}$$

20  $\mathbf{y}_a(\mathbf{x})$ is incorporated into $\mathbf{y}(\mathbf{x})$, and the value of $\mathbf{y}^{mea}$ corresponding to $\mathbf{y}_a(\mathbf{x})$ is zero. The values entered in the weight matrix $\mathbf{W}$ were 0.2 for the real part and 1.25 for the imaginary part. These values are used in the AERONET retrieval (Dubovik and King, 2000) for constraining the spectral variability of the refractive index to some practically reasonable ranges.

In the step 1, $P(\mathbf{x})$ is assumed to be uniform, and the second term of Eq. (5) is ignored. The objective function (Eq. (5)) is

25  minimized by the procedures described in Sects 2.1.4 and 2.1.5, and the columnar properties of the real and imaginary parts of the refractive index ($n^{s1}(\lambda)$ and $k^{s1}(\lambda)$) at SKYR wavelengths, the size distribution ($C_{1,2}^{s1}$, $r_{1,2}^{s1}$, and $s_{1,2}^{s1}$), and the volume ratio of non-spherical particles to total particles in the coarse mode ($\varepsilon^{s1}$) are optimized. The aerosol optical thickness

($\tau_{ext}^{s1}(\lambda)$) and single-scattering albedo ($\omega_0^{s1}(\lambda)$) at SKYR wavelengths are calculated with Equations (6) to (8). $n^{s1}(\lambda)$ and $k^{s1}(\lambda)$ at MIEL wavelengths are calculated by the linear interpolation and extrapolation in the log-log plot. $\tau_{ext}^{s1}(\lambda)$, and $\omega_0^{s1}(\lambda)$ at MIEL wavelengths are calculated from those with Equations (6) to (8). These step 1 results are input to step 2 (see Figs. 1a and b).

**2.1.3 Step 2**

The vertical profiles of the refractive index at MIEL wavelengths, the size distribution, and the volume ratio of the non-spherical particles to total particles in the coarse mode are optimized to MIEL (and HSRL) measurements and the columnar properties obtained in step 1 by the same strategy. The final outputs of the extinction coefficients, single-scattering albedo, and asymmetry factor at MIEL wavelengths are calculated from the optimized aerosol parameters.

The columnar properties input from step 1 are the aerosol optical thickness $\tau_{ext}^{s1}(\lambda)$ and the single-scattering albedo $\omega_0^{s1}(\lambda)$ at MIEL wavelengths. The MIEL and HSRL measurements are three data sets described in Sect. 2.1.1. $\mathbf{y}^{mea}$ for Type 1 data set consists of $\tau_{ext}^{s1}(\lambda)$ and $\omega_0^{s1}(\lambda)$ at MIEL wavelengths, and the vertical profiles of $\beta_{MIE}^{mea}(\lambda, z)$ and $\delta^{mea}(\lambda, z)$ at MIEL wavelengths. $\mathbf{y}^{mea}$ for Type 2 and 3 data sets consist of $\tau_{ext}^{s1}(\lambda)$ and $\omega_0^{s1}(\lambda)$ at MIEL wavelengths, and the vertical profiles of $\beta_{MIE}^{mea}(\lambda, z)$ and $\delta^{mea}(\lambda, z)$ at MIEL wavelengths and $\beta_{RAY}^{mea}(\lambda, z)$ at HSRL wavelengths. The aerosol parameter $\mathbf{x}$ contains the vertical profiles of the real and imaginary parts of the refractive index at MIEL wavelengths ($n(\lambda, z)$ and $k(\lambda, z)$), and the size distribution parameters ($C_1(z)$, $C_2(z)$, and $\varepsilon(z)$). The bi-modal size distribution (Eq. (6)) is also used in step 2, but the mode radii and the widths of the fine and coarse modes are fixed by the columnar values obtained in step 1 ($r_{1,2}^{s1}$, and $s_{1,2}^{s1}$).

In the forward model $\mathbf{y}(\mathbf{x})$ of step 2, the aerosol optical properties at each altitude are calculated with Equations (6) to (8), but note that the extinction/scattering coefficients are calculated.

The normalized attenuated backscatter coefficients for total and molecular scattering are calculated by the lidar equations,

$$\beta_{MIE}(\lambda, z) = \left(\beta_m(\lambda, z) + \beta_p(\lambda, z)\right)$$

$$\exp\left(-2\int_0^z \sigma_{ext,p}\left(\lambda, z^{'}\right) + \sigma_{ext,m}\left(\lambda, z^{'}\right) dz^{'}\right)/\beta_{MIE,ave}(\lambda), \tag{13}$$

$$\beta_{RAY}(\lambda, z) = \beta_m(\lambda, z)\exp\left(-2\int_0^z \sigma_{ext,p}\left(\lambda, z^{'}\right) + \sigma_{ext,m}\left(\lambda, z^{'}\right) dz^{'}\right)/\beta_{RAY,ave}(\lambda), \tag{14}$$

where $\sigma_{ext,p/m}$ are the extinction coefficients for particulate ($p$) and molecular ($m$) scattering, and $\beta_{MIE,ave}(\lambda)$ and $\beta_{RAY,ave}(\lambda)$ are the vertical means of the calculated attenuated backscatter coefficients. The total depolarization ratio ($\delta(\lambda,z)$) at each altitude is calculated with Eqs. (9) to (11).

The first term of Eq. (5) is calculated with the above-mentioned $\mathbf{y}^{mea}$ and $\mathbf{y}(\mathbf{x})$. The values of the diagonal matrix $\mathbf{W}$ were assumed to be 0.01 for $\tau_{ext}^{s1}(\lambda \geq 532nm)$, 0.02 for $\tau_{ext}^{s1}(\lambda < 532nm)$, 0.05 for $\omega_0^{s1}(\lambda)$, 10 % for $\beta_{MIE}^{mea}(\lambda,z)$, 15 % for $\beta_{RAY}^{mea}(\lambda,z)$, and 20 % for $\delta^{mea}(\lambda,z)$.

In step 2, the number of estimated parameters is larger than the number of measurements, so the lidar measurements would be insufficient for retrieving unique solutions of the refractive index even if the columnar properties obtained in step 1 are added to $\mathbf{y}^{mea}$. Therefore, we gave a priori constraints to the real and imaginary parts of the refractive index using the step 1 results $k^{s1}(\lambda)$ and $\tau_{ext}^{s1}(\lambda)$. Assuming the log-normal PDF for $P(\mathbf{x})$, the second term of Eq. (5) is replaced by

$$\frac{1}{2}(\ln\boldsymbol{n} - \ln\boldsymbol{n}_a^{s1})^T (\mathbf{W}_a^2)^{-1}(\ln\boldsymbol{n} - \ln\boldsymbol{n}_a^{s1}) + \frac{1}{2}(\ln\boldsymbol{k} - \ln\boldsymbol{k}_a^{s1})^T (\mathbf{W}_a^2)^{-1}(\ln\boldsymbol{k} - \ln\boldsymbol{k}_a^{s1}). \tag{15}$$

[revised manuscript text omitted]

$$x = \frac{x_{min} + x_{max}\exp(X)}{1 + \exp(X)}.$$ (21)

The value of $x$ is sometimes limited for physical or numerical reasons. In this study, for rapid computation, we constructed the look-up tables of $K^{S/NS}$ in Eqs. (7) and (8) with minimum and maximum values of the refractive index, 1.33 and 1.60 for the real part, and 0.0005 and 0.5 for the imaginary part, respectively. If the refractive index exceeds its maximum or

20    minimum values in the iteration of $\mathbf{x}_{i+1} = \mathbf{x}_i + \alpha_j \mathbf{d}_i$, then the optical properties cannot be calculated and the retrieval process stops. The volume ratio of the non-spherical particles to total particles in the coarse mode must be between 0 and 1 for a physical reason. When the volume ratio of the non-spherical particles has negative values, the values of the size distribution may be negative. These situations can happen due to the large noise of the lidar measurements. The transformation by Eq. (20) prevents $x$ from exceeding its minimum or maximum limitations. Furthermore, because X and Y

[revised manuscript text omitted]

Figures 4 and 5 illustrate the retrieval results from the simulated data for the continental average aerosol with an aerosol optical thickness of 0.05 at 500nm. The "true" values of the refractive index for the externally mixed aerosols were calculated as the weighted arithmetic mean with the scattering coefficient of each aerosol component. The retrieval results with and without HSRL data were the almost same. The retrieval errors of the refractive index were small, except for the real part at 355 nm. Although the volume concentrations of the fine and coarse modes were overestimated, the size distributions at two altitudes were reproduced. However, the coarse mode of the size distribution at lower altitude was overestimated. The volume ratio of the non-spherical particles to total particles in the coarse mode was estimated well except for the result for Type 3 data set. The vertical profiles of the extinction coefficients, the single-scattering albedo, and the asymmetry factor were reproduced. There were the bias errors in the retrieved volume concentrations, but their influences to the extinction coefficients were small because the extinction coefficient depends on not the volume but the cross sectional area. The bias errors of the volume concentrations decreased in the case of the larger aerosol optical thickness (not shown).

We also conducted sensitivity tests using simulated data with random errors to investigate the performance of the algorithm under more realistic conditions. The random error distribution was uniform, and the minimum and maximum values of the random errors were ±2 % for direct solar radiation, ±3 % for diffuse sky radiances, ±5 % for the attenuated backscatter coefficient for total scattering, ±10 % for the attenuated backscatter coefficient for molecular scattering, and ±15 % for the total depolarization ratio. Figures 6 and 7 illustrates the retrieval results from the simulated data for the continental average aerosol, but the simulated data includes random errors. The influences of the random errors to the refractive index were inhibited by the a priori constraints to the refractive index in the step 2 retrieval. The volume concentrations of the fine and coarse modes, and the volume ratio of the non-spherical particles were significantly influenced by the random errors. However, the retrieval errors of the extinction coefficients were small by the reason mentioned in the previous paragraph. The retrieval results of the single-scattering albedo and asymmetry factor exhibited large oscillations, so their vertical profiles were not clear. These oscillations are caused by the retrieval errors in the volume concentrations of the fine and coarse modes because the single-scattering albedo and asymmetry factor depend on not only the refractive index but also the shape of the size distribution.

Figures 8 and 9 present the retrieval results from the simulated data for the transported dust with an aerosol optical thickness of 0.5 at 500 nm. The vertical profiles of the refractive index, except the real part at 355 and 532 nm, were well estimated.

The vertical profiles of the volume concentrations, the size distributions, and the volume ratio of the non-spherical particles were also reproduced. As a result, the vertical profiles of the extinction coefficients, the single-scattering albedo, and the asymmetry factor were reproduced well. Figures 10 and 11 are the retrieval results from the simulated data with random errors. There were small oscillations in the vertical profiles of all the retrievals but the results were almost same as those in Figs. 8 and 9.

Figures 12 and 13 show the retrieval results from the simulated data for the transported pollution aerosol with an aerosol optical thickness of 0.3 at 500nm. The vertical profiles of the volume concentrations, the size distributions, and the real part of the refractive index were estimated well, but the vertical profile of the imaginary part of the refractive index was not. The volume ratio of the non-spherical particles was underestimated at the upper altitude where the volume of the coarse mode was very small. In this test, the large values of the imaginary part of the refractive index and the small values of the single-scattering albedo at upper altitudes were important characteristics that were not reproduced even when HSRL data was used in the retrieval. Figures 14 and 15 are the retrieval results from the simulated data with random errors. All the vertical profiles had small oscillations but were almost same as those in Figs. 12 and 13.

Overall in these tests, our algorithm estimated well the vertical profiles of the size distribution. The vertical profiles of the refractive index were also reproduced, but the real parts of the refractive index of the transported dust and the imaginary part of the refractive index of the transported pollution aerosol were not. The vertical profile of the extinction coefficient was reproduced well because the size distribution is a most important factor to determine the value of the extinction coefficient. The vertical profile of the asymmetry factor was also reproduced because the asymmetry factor depends on the shape of the size distribution firstly and the real part of the refractive index secondly. The vertical profile of the single-scattering albedo was reproduced in the case of transported dust because the shape of the size distribution and the imaginary part of the refractive index were reproduced. However, the vertical profiles of the single-scattering albedo and the imaginary part of the refractive index were not in the case of transported pollution aerosol. These characteristics were consistently observed regardless of the aerosol optical thickness value of the simulated data.

To investigate the difference of the retrievals of the refractive index between the transported dust and pollution aerosol, we calculated the dependencies of the extinction-to backscatter ratio on the refractive index and the mode radius (Fig. 16). The mono-modal lognormal size distribution (Eq. 6) with a mode width of 0.5 was used in the calculation. The optical properties of the randomly oriented polydisperse spheroids described in Sect. 2.1.2 were used in the case of the non-spherical particle. The extinction-to backscatter ratios at all the wavelengths were widely changed by the real part of the refractive index in the results of the coarse mode (the mode radius from 1 to 5 μm), but such large variations were not seen in the results of the fine mode (the mode radius from 0.1 to 0.2μm). The extinction-to backscatter ratios depended on the imaginary part of the refractive index in the results of both the fine and coarse modes. However, for the fine mode, the dependencies were small in the limited range of the imaginary part from 0.005 to 0.02. This is a range of our defined transported pollution aerosol (Fig.

12). Consequently, the extinction-to backscatter ratio is sensitive to the real and imaginary parts of the refractive index in the case that the coarse mode is dominant but is not in the case of the fine mode. These features were seen in the results of both the spherical and non-spherical particles.

The retrieval results obtained with and without HSRL data did not differ. The advantage of HSRL data is that particle backscatter and extinction coefficients are obtained separately. Our algorithm in the step 1 also estimates the optical thickness and the other parameters related to the backscatter coefficient separately. The retrievals of the step 1 are the columnar values, but manipulating the MIEL data with the step 1 retrievals would have an effect similar to the addition of HSRL data. In this regard, our algorithm cannot utilize HSRL data; thus further development of the algorithm is necessary.

The influence of the random error was small when the optical thickness was large. However, the random errors had a large influence when the aerosol optical thickness was small (Figs. 6 and 7). In the methods of Chaikovsky et al. (2012) and Lopatin et al. (2013), the smoothness constraints for the vertical profiles of the volumes in the fine and coarse modes are introduced to overcome the random noise of the lidar measurements. Since this smoothness constraint is expected to decrease the large oscillations of 
[revised manuscript text omitted]

|---|---|---|---|---|---|---|---|---|
| Continental average | Water-soluble | Sphere | 0.18 | 0.81 | 1.44 | 0.0026 | 0.90 | $\exp(-z/H)$, H = 8 km |
| | Soot | Sphere | 0.05 | 0.69 | 1.75 | 0.45 | 0.07 | $\exp(-z/H)$, H = 4 km |
| | Insoluble | Spheroid | 5.98 | 0.92 | 1.53 | 0.008 | 0.03 | $\exp(-z/H)$, H = 2 km |
| Transported dust | Dust | Spheroid | 3.23 | 0.79 | 1.53 | 0.0078 | 0.25 | $\frac{1}{\sqrt{2\pi}\sigma}\exp\left(-\frac{(z-z_c)}{2\sigma^2}\right)$, $z_c = 3.5$ km, $\sigma = 0.4$ km |
| | Water-soluble | Sphere | 0.18 | 0.81 | 1.44 | 0.0026 | 0.67 | $\exp(-z/H)$, H = 8 km |
| | Soot | Sphere | 0.05 | 0.69 | 1.75 | 0.45 | 0.05 | $\exp(-z/H)$, H = 4 km |
| | Insoluble | Spheroid | 5.98 | 0.92 | 1.53 | 0.008 | 0.03 | $\exp(-z/H)$, H = 2 km |
| Transported pollution | Water-soluble | Sphere | 0.18 | 0.81 | 1.44 | 0.0026 | 0.08 | $\frac{1}{\sqrt{2\pi}\sigma}\exp\left(-\frac{(z-z_c)}{2\sigma^2}\right)$, $z_c = 3.5$ km, $\sigma = 0.4$ km |
| | Soot | Sphere | 0.05 | 0.69 | 1.75 | 0.45 | 0.03 | $\frac{1}{\sqrt{2\pi}\sigma}\exp\left(-\frac{(z-z_c)}{2\sigma^2}\right)$, $z_c = 3.5$ km, $\sigma = 0.4$ km |
| | Water-soluble | Sphere | 0.18 | 0.81 | 1.44 | 0.0026 | 0.79 | $\exp(-z/H)$, H = 8 km |
| | Soot | Sphere | 0.05 | 0.69 | 1.75 | 0.45 | 0.06 | $\exp(-z/H)$, H = 4 km |
| | Insoluble | Spheroid | 5.98 | 0.92 | 1.53 | 0.008 | 0.03 | $\exp(-z/H)$, H = 2 km |

**(a) Step 1: Columnar property**

[Figure]

**(b) Step 2: Vertical profile**

[Figure]

**(c) Update **x**

[revised manuscript text omitted]

---

## Author Comment (AC3) · 17 Jun 2016

[revised manuscript text omitted]

Our algorithm estimates the aerosol parameters, based on the maximum a posteriori (MAP) scheme. The aerosol parameters for the best fit to all of the measurements and a priori information are obtained by maximizing the posterior probability density function (PDF):

where vector  $\mathbf{y}^{mea}$  describes the measurements, vector  $\mathbf{x}$  describes the aerosol parameters to be estimated,  $P(\mathbf{y}^{mea}|\mathbf{x})$  is the conditional PDF of  $\mathbf{y}^{mea}$  given  $\mathbf{x}$ ,  $P(\mathbf{x})$  is the prior PDF of  $\mathbf{x}$ , and  $P(\mathbf{y}^{mea})$  is the prior PDF of  $\mathbf{y}^{mea}$ . Since  $P(\mathbf{y}^{mea})$  does not depend on  $\mathbf{x}$ , the PDF to be maximized is  $P(\mathbf{y}^{mea}|\mathbf{x})P(\mathbf{x})$ . Assuming the normal distribution,  $P(\mathbf{y}^{mea}|\mathbf{x})P(\mathbf{x})$  is defined as

$$P(\mathbf{y}^{mea}|\mathbf{x})P(\mathbf{x}) \propto \exp\left[-\frac{1}{2}\left(\mathbf{y}^{mea} - \mathbf{y}(\mathbf{x})\right)^{T}(\mathbf{W}^{2})^{-1}\left(\mathbf{y}^{mea} - \mathbf{y}(\mathbf{x})\right)\right]$$
$$-\frac{1}{2}(\mathbf{x} - \mathbf{x}_{a})^{T}(\mathbf{W}_{a}^{2})^{-1}(\mathbf{x} - \mathbf{x}_{a})], \tag{4}$$

10

5

where vector  $\mathbf{y}(\mathbf{x})$  comprise the values corresponding to  $\mathbf{y}^{mea}$  calculated from  $\mathbf{x}$  by the forward model, and matrix  $\mathbf{W}^2$  is the covariance matrix of  $\mathbf{y}$  and is assumed to be diagonal in this study. The diagonal elements of  $\mathbf{W}$  are the standard errors in the measurements. The vector  $\mathbf{x}_a$  comprises the a priori value of  $\mathbf{x}$ , and matrix  $\mathbf{W}_a$  is the associated covariance matrix. The maximum of Eq. (4) is obtained by minimizing the objective function,

$$f(\mathbf{x}) = \frac{1}{2} \left( \mathbf{y}^{mea} - \mathbf{y}(\mathbf{x}) \right)^T \left( \mathbf{W}^2 \right)^{-1} \left( \mathbf{y}^{mea} - \mathbf{y}(\mathbf{x}) \right) + \frac{1}{2} \left( \mathbf{x} - \mathbf{x}_a \right)^T \left( \mathbf{W}_a^2 \right)^{-1} \left( \mathbf{x} - \mathbf{x}_a \right).$$
(5)

15

20

We search for the best **x**, which minimizes  $f(\mathbf{x})$ , by iterations of the Gauss-Newton method with a line search,  $\mathbf{x}_{i+1} = \mathbf{x}_i + \alpha_i \mathbf{d}_i$  ("Update **x**" in Fig. 1). This minimization procedure is described in Sects 2.1.4 and 2.1.5.

In step 1, the vector  $\mathbf{y}^{mea}$  consists of  $\tau_{ext}^{mea}(\lambda)$  at SKYR wavelengths,  $I^{mea}(\Theta, \lambda)$  at SKYR wavelengths and scattering angles, and  $\delta_{ave}^{mea}$  at 532 nm. Vector  $\mathbf{x}$  describes the aerosol parameters and consists of the real and imaginary parts of the refractive index at SKYR wavelengths  $(n(\lambda) \text{ and } k(\lambda))$ , and the parameters for the size distribution  $(C_1, C_2, r_{m,1}, r_{m,2}, s_1, s_2,$ and  $\varepsilon$ ). We assumed the following bi-modal lognormal size distribution,

$$\frac{dV(r)}{d\ln r} = \sum_{i=1}^{2} \frac{dV_i(r)}{d\ln r} = \sum_{i=1}^{2} \frac{C_i}{\sqrt{2\pi s_i}} \exp\left[-\frac{1}{2} \left(\frac{\ln r - \ln r_{m,i}}{s_i}\right)^2\right],$$

(6)

where  $C_i$ ,  $r_{m,i}$ , and  $s_i$  are volume, radius, and width, respectively of the fine (i = 1) and coarse (i = 2) modes.  $\varepsilon$  is the volume ratio of non-spherical particles to total particles in the coarse mode.

We constructed the forward models  $\mathbf{y}(\mathbf{x})$  to calculate  $\tau_{ext}(\lambda)$ ,  $I(\Theta, \lambda)$ , and  $\delta_{ave}(532\text{ nm})$  from the above-mentioned aerosol parameters. The optical properties of aerosols were calculated by a method similar to that of **Dubovik et al.** (2006) as follows:

$$\tau_{\frac{ext}{sca}}(\lambda) = \sum_{k} \frac{dV_{1}(r_{k})}{d\ln r} K_{\frac{ext}{sca}}^{S}(\lambda, n, k, r_{k}) + \sum_{k} (1 - \varepsilon) \frac{dV_{2}(r_{k})}{d\ln r} K_{\frac{ext}{sca}}^{S}(\lambda, n, k, r_{k}) + \sum_{k} \varepsilon \frac{dV_{2}(r_{k})}{d\ln r} K_{\frac{ext}{sca}}^{NS}(\lambda, n, k, r_{k}),$$
(7)

10
$$\tau_{sca}(\lambda)P_{ii}(\Theta,\lambda) = \sum_{k} \frac{dV_{1}(r_{k})}{d\ln r} K_{ii}^{S}(\Theta,\lambda,n,k,r_{k}) + \sum_{k} \varepsilon \frac{dV_{2}(r_{k})}{d\ln r} K_{ii}^{NS}(\Theta,\lambda,n,k,r_{k}), \qquad (8)$$

where  $\tau_{ext/sca}(\lambda)$  denotes the optical thickness for extinction and scattering, and  $\tau_{sca}(\lambda)P_{ii}(\theta,\lambda)$  denotes the directional scattering corresponding to the scattering matrix elements  $P_{ii}(\theta,\lambda)$ .  $K_{...}^{S}$  and  $K_{...}^{NS}$  are the kernels of extinction and scattering properties for spherical and non-spherical particles, respectively. The kernel for spherical particles was constructed by using Mie theory. The kernel for non-spherical particles was constructed by using the data table of Dubovik et al. (2006), which assumes randomly oriented polydisperse spheroids with a fixed aspect ratio distribution for mineral dust.

 $I(\theta, \lambda)$  is computed by the radiative transfer code in the SKYRAD.PACK ver. 4.2 (Nakajima et al., 1996). Although only one atmospheric layer is considered by SKYRAD.PACK, we assumed that the atmosphere consists of two layers. The bottom layer includes aerosols, and its top altitude  $z_{max}$  is determined by lidar measurement. The upper layer is aerosol-free. The Rayleigh scattering is calculated by the method of Bucholtz (1995). The vertical ozone profile is approximated by the formula of Green (1964), and the ozone absorption coefficient is adopted from the LOWTRAN 7 database (Kneizys et al., 1988).

 $\delta_{ave}(\lambda)$  is calculated as

25

15

$$\delta_{ave}(\lambda) = \left(\frac{\beta_m(\lambda)\delta_m(\lambda)}{1+\delta_m(\lambda)} + \frac{\beta_p(\lambda)\delta_p(\lambda)}{1+\delta_p(\lambda)}\right) / \left(\frac{\beta_m(\lambda)}{1+\delta_m(\lambda)} + \frac{\beta_p(\lambda)}{1+\delta_p(\lambda)}\right),$$

$$\beta_{p/m}(\lambda) = \tau_{sca,p/m}(\lambda) P_{11,p/m}(180^{\circ},\lambda),$$

$$\delta_{p/m}(\lambda) = \frac{1 - P_{22,p/m}(180°,\lambda)/P_{11,p/m}(180°,\lambda)}{1 + P_{22,p/m}(180°,\lambda)/P_{11,p/m}(180°,\lambda)},\tag{11}$$

where  $\beta_{p/m}(\lambda)$ ,  $\delta_{p/m}(\lambda)$ , and  $\tau_{sca,p/m}(\lambda)$  are the backscatter coefficient, the depolarization ratio, and the scattering optical thickness, respectively, for particulate (*p*) and molecular (*m*) scattering in the bottom aerosol layer.

The first term of Eq. (5) is calculated by using the above-mentioned  $\mathbf{y}^{mea}$  and  $\mathbf{y}(\mathbf{x})$ . The values of the diagonal matrix  $\mathbf{W}$  are the measurement errors, which were assumed to be 0.01 for  $\tau_{ext}^{mea}$  ( $\lambda \ge 500$ nm) and 0.02 for  $\tau_{ext}^{mea}$  ( $\lambda < 500$ nm), 5 % for  $I^{mea}(\Theta, \lambda)$ , and 20 % for  $\delta_{ave}^{mea}$  (532 nm).

10

To reduce the effects of measurement errors on retrievals, Dubovik and King (2000) restricted the spectral variability of the refractive index by limiting the length of the derivatives of the refractive index with respect to the wavelength. They considered this a priori smoothness constraint as being of the same nature as a measurement and incorporated the smoothness constraint into their retrieval scheme. We adopted this consideration and introduced the smoothness constraint

15 for the real and imaginary parts of the refractive index. The first derivatives of the refractive index with respect to the wavelengths are defined as

$$\mathbf{y}_{a}(\mathbf{x}) = \left(\frac{\ln n(\lambda_{1}) - \ln n(\lambda_{2})}{\ln \lambda_{1} - \ln \lambda_{2}} \cdots \frac{\ln n(\lambda_{6}) - \ln n(\lambda_{7})}{\ln \lambda_{6} - \ln \lambda_{7}} \frac{\ln k(\lambda_{1}) - \ln k(\lambda_{2})}{\ln \lambda_{1} - \ln \lambda_{2}} \cdots \frac{\ln k(\lambda_{6}) - \ln k(\lambda_{7})}{\ln \lambda_{6} - \ln \lambda_{7}}\right).$$
(12)

20  $\mathbf{y}_{\mathbf{a}}(\mathbf{x})$  is incorporated into  $\mathbf{y}(\mathbf{x})$ , and the value of  $\mathbf{y}^{mea}$  corresponding to  $\mathbf{y}_{\mathbf{a}}(\mathbf{x})$  is zero. The values entered in the weight matrix  $\mathbf{W}$  were 0.2 for the real part and 1.25 for the imaginary part. These values are used in the AERONET retrieval (Dubovik and King, 2000) for constraining the spectral variability of the refractive index to some practically reasonable ranges.

In the step 1,  $P(\mathbf{x})$  is assumed to be uniform, and the second term of Eq. (5) is ignored. The objective function (Eq. (5)) is 25 minimized by the procedures described in Sects 2.1.4 and 2.1.5, and the columnar properties of the real and imaginary parts of the refractive index  $(n^{s_1}(\lambda) \text{ and } k^{s_1}(\lambda))$  at SKYR wavelengths, the size distribution  $(C_{1,2}^{s_1}, r_{1,2}^{s_1}, \text{ and } s_{1,2}^{s_1})$ , and the volume ratio of non-spherical particles to total particles in the coarse mode  $(\varepsilon^{s_1})$  are optimized. The aerosol optical thickness

(<mark>10</mark>)

 $(\tau_{ext}^{s1}(\lambda))$  and single-scattering albedo  $(\omega_0^{s1}(\lambda))$  at SKYR wavelengths are calculated with Equations (6) to (8).  $n^{s1}(\lambda)$  and  $k^{s1}(\lambda)$  at MIEL wavelengths are calculated by the linear interpolation and extrapolation in the log-log plot.  $\tau_{ext}^{s1}(\lambda)$ , and  $\omega_0^{s1}(\lambda)$  at MIEL wavelengths are calculated from those with Equations (6) to (8). These step 1 results are input to step 2 (see Figs. 1a and b).

**5 2.1.3 Step 2**

The vertical profiles of the refractive index at MIEL wavelengths, the size distribution, and the volume ratio of the nonspherical particles to total particles in the coarse mode are optimized to MIEL (and HSRL) measurements and the columnar properties obtained in step 1 by the same strategy. The final outputs of the extinction coefficients, single-scattering albedo, and asymmetry factor at MIEL wavelengths are calculated from the optimized aerosol parameters.

- 10 The columnar properties input from step 1 are the aerosol optical thickness  $\tau_{ext}^{s1}(\lambda)$  and the single-scattering albedo  $\omega_0^{s1}(\lambda)$  at MIEL wavelengths. The MIEL and HSRL measurements are three data sets described in Sect. 2.1.1.  $\mathbf{y}^{mea}$  for Type 1 data set consists of  $\tau_{ext}^{s1}(\lambda)$  and  $\omega_0^{s1}(\lambda)$  at MIEL wavelengths, and the vertical profiles of  $\beta_{MIE}^{mea}(\lambda, z)$  and  $\delta^{mea}(\lambda, z)$  at MIEL wavelengths, and the vertical profiles of  $\beta_{MIE}^{mea}(\lambda, z)$  and  $\delta^{mea}(\lambda, z)$  at MIEL wavelengths of  $\sigma_{RAY}^{s1}(\lambda)$  at MIEL wavelengths. The aerosol parameter  $\mathbf{x}$  contains
- the vertical profiles of the real and imaginary parts of the refractive index at MIEL wavelengths (n(λ, z) and k(λ, z)), and the size distribution parameters (C1(z), C2(z), and ε(z)). The bi-modal size distribution (Eq. (6)) is also used in step 2, but the mode radii and the widths of the fine and coarse modes are fixed by the columnar values obtained in step 1 (rs11,2, and ss11,2). In the forward model y(x) of step 2, the aerosol optical properties at each altitude are calculated with Equations (6) to (8),
- 20 The normalized attenuated backscatter coefficients for total and molecular scattering are calculated by the lidar equations,

but note that the extinction/scattering coefficients are calculated.

$$\beta_{MIE}(\lambda, z) = \left(\beta_m(\lambda, z) + \beta_p(\lambda, z)\right)$$
$$\exp\left(-2\int_0^z \sigma_{ext,p}\left(\lambda, z^{\prime}\right) + \sigma_{ext,m}\left(\lambda, z^{\prime}\right) dz^{\prime}\right) / \beta_{MIE,ave}(\lambda),$$
(13)

25
$$\beta_{RAY}(\lambda, z) = \beta_m(\lambda, z) \exp\left(-2\int_0^z \sigma_{ext,p}\left(\lambda, z^{\prime}\right) + \sigma_{ext,m}\left(\lambda, z^{\prime}\right) dz^{\prime}\right) / \beta_{RAY,ave}(\lambda),$$
(14)

where  $\sigma_{ext,p/m}$  are the extinction coefficients for particulate (p) and molecular (m) scattering, and  $\beta_{MIE,ave}(\lambda)$  and  $\beta_{RAY,aye}(\lambda)$  are the vertical means of the calculated attenuated backscatter coefficients. The total depolarization ratio  $(\delta(\lambda, z))$  at each altitude is calculated with Eqs. (9) to (11).

The first term of Eq. (5) is calculated with the above-mentioned  $\mathbf{y}^{mea}$  and  $\mathbf{y}(\mathbf{x})$ . The values of the diagonal matrix  $\mathbf{W}$  were assumed to be 0.01 for  $\tau_{ext}^{s1}$  ( $\lambda \ge 532nm$ ), 0.02 for  $\tau_{ext}^{s1}$  ( $\lambda < 532nm$ ), 0.05 for  $\omega_0^{s1}(\lambda)$ , 10 % for  $\beta_{MIE}^{mea}(\lambda, z)$ , 15 % for 5  $\beta_{RAY}^{mea}(\lambda, z)$ , and 20 % for  $\delta^{mea}(\lambda, z)$ .

In step 2, the number of estimated parameters is larger than the number of measurements, so the lidar measurements would be insufficient for retrieving unique solutions of the refractive index even if the columnar properties obtained in step 1 are added to  $y^{mea}$ . Therefore, we gave a priori constraints to the real and imaginary parts of the refractive index using the step 1 results  $k^{s1}(\lambda)$  and  $\tau^{s1}_{ext}(\lambda)$ . Assuming the log-normal PDF for  $P(\mathbf{x})$ , the second term of Eq. (5) is replaced by

 $\frac{1}{2}(\ln \boldsymbol{n} - \ln \boldsymbol{n}_a^{s1})^T (\mathbf{W}_a^2)^{-1} (\ln \boldsymbol{n} - \ln \boldsymbol{n}_a^{s1}) + \frac{1}{2}(\ln \boldsymbol{k} - \ln \boldsymbol{k}_a^{s1})^T (\mathbf{W}_a^2)^{-1} (\ln \boldsymbol{k} - \ln \boldsymbol{k}_a^{s1}).$

The matrix  $\mathbf{W}_{a}$ , which determines the strength of the constraints, is assumed to be diagonal and the values are obtained by a method similar to that of Dubovik and King (2000). The possible variability ranges of the refractive index for aerosols are from 1.33 to 1.6 for the real part, and from 0.0005 to 0.5 for the imaginary part. We considered these intervals to be 68 % confidence intervals,  $[\ln n^{s_1} - \Delta n, \ln n^{s_1} + \Delta n]$  and  $[\ln k^{s_1} - \Delta k, \ln k^{s_1} + \Delta k]$ , and determined the weight values as

(15)

$$W_{a} = \begin{cases} 0.5(\ln n_{max} - \ln n_{min}), & \text{for real part} \\ 0.5(\ln k_{max} - \ln k_{min}), & \text{for imaginary part} \end{cases}$$
(16)

20

10

15

The objective function is minimized by the procedures described in Sects 2.1.4 and 2.1.5, and the vertical profiles of the real and imaginary parts of the refractive index  $(n(\lambda, z), k(\lambda, z))$  at MIEL wavelengths, the size distribution  $(C_1(z), C_2(z))$ , and the volume ratio of non-spherical particles to total particles in the coarse mode ( $\varepsilon(z)$ ) are optimized. Finally, the vertical profiles of the refractive index, size distribution  $(\frac{dV(r,z)}{d\ln r})$ , extinction coefficients  $(\sigma_{ext}(\lambda, z))$ , single-scattering albedo  $(\omega_0(\lambda, z))$ , and asymmetry parameter  $(q(\lambda, z))$  are produced as output. The wavelengths of these optical properties are 532 and 1064 nm for Type 1 and 2 data sets, and 355, 532, 1064 nm for Type 3 data set.

**2.1.4 Minimization procedure**

5

15

In both steps 1 and 2, **x** was optimized to  $\mathbf{y}^{mea}$  by finding the minimum of  $f(\mathbf{x})$  in **x** space. We employed the Gauss-Newton method to search for the minimum. The Gauss-Newton method searches for the minimum by iteratively updating **x**, and it is usually combined with a line search method (Nocedal and Wright, 2006). In this procedure (Fig. 1c), **x** is updated by  $\mathbf{x}_{i+1} = \mathbf{x}_i + \alpha_j \mathbf{d}_i$ , where the value of  $f(\mathbf{x})$  in the **x** space decreases in the vector  $\mathbf{d}_i$  direction, and  $\alpha_j$  is a positive parameter that minimizes  $f(\mathbf{x})$  in direction  $\mathbf{d}_i$ .  $\mathbf{d}_i$  is determined by the Gauss-Newton method in the outer-loop, and  $\alpha_j$  is determined by a line search in the inner loop.  $\mathbf{d}_i$  is obtained by solving the normal equation,

$$[\mathbf{J}(\mathbf{x}_{i})^{T}(\mathbf{w}^{2})^{-1}\mathbf{J}(\mathbf{x}_{i}) + (\mathbf{w}_{a}^{2})^{-1}]\mathbf{d}_{i} = \mathbf{J}(\mathbf{x}_{i})^{T}(\mathbf{w}^{2})^{-1}(\mathbf{y}^{\text{mea}} - \mathbf{y}(\mathbf{x}_{i}))$$

$$10 + (\mathbf{w}_{a}^{2})^{-1}(\mathbf{x}_{i} - \mathbf{x}_{a}), \qquad (17)$$

where  $\mathbf{J}(\mathbf{x}_i)$  is the Jacobi matrix and is calculated as the first derivatives of  $\mathbf{y}(\mathbf{x}_i)$  in the near vicinity of  $\mathbf{x}_i$ . We solved this normal equation by Singular Value Decomposition (Press et al., 1992). After  $\mathbf{d}_i$  is determined,  $\alpha_j$  is searched for by the iteration of  $\alpha_{j+1} = \eta \alpha_j$ . The initial value of  $\alpha_j$  is 1.0, and the value of  $\eta$  is set to 0.5.  $\alpha_j$  is iteratively decreased until the Armijo condition is satisfied,

$$f(\mathbf{x}_i + \alpha_i \mathbf{d}_i) \le f(\mathbf{x}_i) + \gamma \alpha_i \nabla f(\mathbf{x}_i)^T \mathbf{d}_i, 0 < \gamma < 1,$$
(18)

[revised manuscript text omitted]

- Figures 4 and 5 illustrate the retrieval results from the simulated data for the continental average aerosol with an aerosol optical thickness of 0.05 at 500nm. The "true" values of the refractive index for the externally mixed aerosols were calculated as the weighted arithmetic mean with the scattering coefficient of each aerosol component. The retrieval results with and without HSRL data were the almost same. The retrieval errors of the refractive index were small, except for the real
- part at 355 nm. Although the volume concentrations of the fine and coarse modes were overestimated, the size distributions at two altitudes were reproduced. However, the coarse mode of the size distribution at lower altitude was overestimated. The volume ratio of the non-spherical particles to total particles in the coarse mode was estimated well except for the result for Type 3 data set. The vertical profiles of the extinction coefficients, the single-scattering albedo, and the asymmetry factor were reproduced. There were the bias errors in the retrieved volume concentrations, but their influences to the extinction coefficient depends on not the volume but the cross sectional area. The bias
  - errors of the volume concentrations decreased in the case of the larger aerosol optical thickness (not shown).

5

We also conducted sensitivity tests using simulated data with random errors to investigate the performance of the algorithm under more realistic conditions. The random error distribution was uniform, and the minimum and maximum values of the random errors were ±2 % for direct solar radiation, ±3 % for diffuse sky radiances, ±5 % for the attenuated backscatter coefficient for total scattering, ±10 % for the attenuated backscatter coefficient for molecular scattering, and ±15 % for the total depolarization ratio. Figures 6 and 7 illustrates the retrieval results from the simulated data for the continental average aerosol, but the simulated data includes random errors. The influences of the random errors to the refractive index were inhibited by the a priori constraints to the refractive index in the step 2 retrieval. The volume concentrations of the fine and coarse modes, and the volume ratio of the non-spherical particles were significantly influenced by the random errors.
25 However, the retrieval errors of the extinction coefficients were small by the reason mentioned in the previous paragraph. The retrieval results of the single-scattering albedo and asymmetry factor exhibited large oscillations, so their vertical

- profiles were not clear. These oscillations are caused by the retrieval errors in the volume concentrations of the fine and coarse modes because the single-scattering albedo and asymmetry factor depend on not only the refractive index but also the shape of the size distribution.
- 30 Figures 8 and 9 present the retrieval results from the simulated data for the transported dust with an aerosol optical thickness of 0.5 at 500 nm. The vertical profiles of the refractive index, except the real part at 355 and 532 nm, were well estimated.

The vertical profiles of the volume concentrations, the size distributions, and the volume ratio of the non-spherical particles were also reproduced. As a result, the vertical profiles of the extinction coefficients, the single-scattering albedo, and the asymmetry factor were reproduced well. Figures 10 and 11 are the retrieval results from the simulated data with random errors. There were small oscillations in the vertical profiles of all the retrievals but the results were almost same as those in

5 **Figs. 8 and 9**.

Figures 12 and 13 show the retrieval results from the simulated data for the transported pollution aerosol with an aerosol optical thickness of 0.3 at 500nm. The vertical profiles of the volume concentrations, the size distributions, and the real part of the refractive index were estimated well, but the vertical profile of the imaginary part of the refractive index was not. The volume ratio of the non-spherical particles was underestimated at the upper altitude where the volume of the coarse mode

10 was very small. In this test, the large values of the imaginary part of the refractive index and the small values of the single-scattering albedo at upper altitudes were important characteristics that were not reproduced even when HSRL data was used in the retrieval. Figures 14 and 15 are the retrieval results from the simulated data with random errors. All the vertical profiles had small oscillations but were almost same as those in Figs. 12 and 13.

Overall in these tests, our algorithm estimated well the vertical profiles of the size distribution. The vertical profiles of the 15 refractive index were also reproduced, but the real parts of the refractive index of the transported dust and the imaginary part of the refractive index of the transported pollution aerosol were not. The vertical profile of the extinction coefficient was reproduced well because the size distribution is a most important factor to determine the value of the extinction coefficient. The vertical profile of the asymmetry factor was also reproduced because the asymmetry factor depends on the shape of the size distribution firstly and the real part of the refractive index secondly. The vertical profile of the single-scattering albedo 20 was reproduced in the case of transported dust because the shape of the size distribution and the imaginary part of the refractive index were reproduced. However, the vertical profiles of the single-scattering albedo and the imaginary part of the refractive index were not in the case of transported pollution aerosol. These characteristics were consistently observed

regardless of the aerosol optical thickness value of the simulated data.

To investigate the difference of the retrievals of the refractive index between the transported dust and pollution aerosol, we calculated the dependencies of the extinction-to backscatter ratio on the refractive index and the mode radius (Fig. 16). The mono-modal lognormal size distribution (Eq. 6) with a mode width of 0.5 was used in the calculation. The optical properties of the randomly oriented polydisperse spheroids described in Sect. 2.1.2 were used in the case of the non-spherical particle. The extinction-to backscatter ratios at all the wavelengths were widely changed by the real part of the refractive index in the results of the coarse mode (the mode radius from 1 to 5 µm), but such large variations were not seen in the results of the fine 30 mode (the mode radius from 0.1 to 0.2µm). The extinction-to backscatter ratios depended on the imaginary part of the refractive index in the refractive index in the results of both the fine and coarse modes. However, for the fine mode, the dependencies were small in the limited range of the imaginary part from 0.005 to 0.02. This is a range of our defined transported pollution aerosol (Fig.

12). Consequently, the extinction-to backscatter ratio is sensitive to the real and imaginary parts of the refractive index in the case that the coarse mode is dominant but is not in the case of the fine mode. These features were seen in the results of both the spherical and non-spherical particles.

- The retrieval results obtained with and without HSRL data did not differ. The advantage of HSRL data is that particle backscatter and extinction coefficients are obtained separately. Our algorithm in the step 1 also estimates the optical thickness and the other parameters related to the backscatter coefficient separately. The retrievals of the step 1 are the columnar values, but manipulating the MIEL data with the step 1 retrievals would have an effect similar to the addition of HSRL data. In this regard, our algorithm cannot utilize HSRL data; thus further development of the algorithm is necessary.
- The influence of the random error was small when the optical thickness was large. However, the random errors had a large influence when the aerosol optical thickness was small (Figs. 6 and 7). In the methods of Chaikovsky et al. (2012) and Lopatin et al. (2013), the smoothness constraints for the vertical profiles of the volumes in the fine and coarse modes are introduced to overcome the random noise of the lidar measurements. Since this smoothness constraint is expected to decrease the large oscillations of 
[revised manuscript text omitted]
 optical thickness, the single-scattering albedo, and asymmetry factor from the SKYR measurements. The most different

- 20 the optical thickness, the single-scattering albedo, and asymmetry factor from the SKYR measurements. The most different point from the step 1 of the SKYLIDAR algorithm is the assumptions of the size distribution. In the SKYRAD.PACK, only the spherical particle is assumed, and the size distribution consists of the 20 discrete bins. In the SKYLIDAR algorithm, the non-spherical particle is considered, and the bi-modal size distribution is used.
- The aerosol optical thickness at 532 and 1064 nm in the SKYLIDAR retrievals agreed well overall with those of the SKYRAD.PACK retrievals. Although slightly underestimated, the SKYLIDAR single-scattering albedo at 532 nm agreed well with the SKYRAD.PACK retrieval for aerosol optical thickness of more than 0.2. The single-scattering albedo estimated from the SKYR measurements by SKYRAD.PACK, however, is larger than that of the AERONET retrievals (Che et al. 2008). Thus, the SKYLIDAR results may be close to the AERONET retrievals. Similarly, the asymmetry factor also agreed with those estimated by SKYRAD.PACK for aerosol optical thickness of more than 0.2. Comparison of the two-year
- 30 mean of the normalized volume size distribution showed agreement with respect to the fine mode but not the coarse mode;

[revised manuscript text omitted]

irradiance.

We have a plan to analyze the long-term measurements of SKYR (SKYNET) and MIEL (AD-Net) in the East Asian region using the SKYLIDAR algorithm. The results of the aerosol vertical profile and the heating rate would reveal the characteristics of the locally emitted and transported aerosols, and their influences to the temperature profiles. We focused

- 20 on the measurements of SKYNET and AD-Net, but the SKYLIDAR algorithm can be applied to an another data set similar to **SKYR** and MIEL measurements. This flexibility is expected to be useful for investigating the temporal and spatial distribution of aerosols at different observational sites. In addition, the minimization procedure with our developed logarithm transformations, which works well for hundreds of estimated parameters, is useful for the various remote sensing. Using this minimization techniques, we are now developing the synergetic methods to estimate the aerosol vertical profile from the 25 space-borne lidar and imager data, a combination of CALIOP and MODIS, and a combination of ATLID and MSI/

EarthCARE (Illingworth et al. 2015).

**Acknowledgements**

This work was supported by the Japan Society for the Promotion of Science KAKENHI Grant No. 24510026, 15H01728, and 15H02808. The authors are grateful to the OpenCLASTR project for allowing us to use the SKYRAD.PACK (sky radiometer analysis package) in this research. NCEP reanalysis data were provided by the NOAA/OAR/ESRL PSD, Boulder, Colorado, USA, from its Web site at http://www.esrl.noaa.gov/psd/. The MODIS MCD12C1 product was retrieved from the online Data Pool, courtesy of the NASA EOSDIS Land Processes Distributed Active Archive Center (LP DAAC), USGS/Earth Resources Observation and Science (EROS) Center, Sioux Falls, South Dakota, https://lpdaac.usgs.gov/data access/data pool.

[revised manuscript text omitted]

**(a) Step 1: Columnar property**

(b) Step 2: Vertical profile